

# Modeling stable and unstable flow in unsaturated porous media for different infiltration rates

Jakub Kmec[1,2] and Miloslav Šír[1]

[1]Palacký University Olomouc, Faculty of Science, Joint Laboratory of Optics of Palacký University and Institute of Physics of the Czech Academy of Sciences, 17. listopadu 1192/12, 779 00 Olomouc, Czech Republic
[2]Institute of Physics of the Czech Academy of Sciences, Joint Laboratory of Optics of Palacký University and Institute of Physics of the Czech Academy of Sciences, 17. listopadu 1154/50a, 779 00 Olomouc, Czech Republic

**Correspondence:** Jakub Kmec (jakub.kmec@upol.cz)

**Abstract.** The gravity-driven flow in unsaturated porous medium is still one of the biggest unsolved problems in multiphase flow. Sometimes a stable flow with an uniform wetting front is observed, but at other times it is unstable with distinct preferential pathways even if the porous material is homogeneous. The formation of an unstable wetting front in a porous medium depends on many factors such as the type of the porous medium, the initial saturation or the applied infiltration rate. As the
infiltration rate increases, the wetting front first transitions from stable to unstable for low infiltration rates, and then from unstable to stable for high infiltration rates. We propose a governing equation and its discretized form, the semi-continuum model, to describe this significant non-monotonic transition. We show that the semi-continuum model is able to capture the influx dependence together with the correct finger width and spacing. We also present that the instability of the wetting front is closely related to the saturation overshoot in 1D. Finally, we demonstrate that the flow can be still preferential even when the
porous medium is completely wetted.

## 1 Introduction

In hydrology, the gravity-driven multiphase flow in porous media is a long-standing and still unsolved problem. Typically, this involves the flow of water into the soil, which is extremely complicated physical phenomenon and exhibits counter-intuitive
behavior. The standard concept of the diffusion-like flow does not apply here (DiCarlo, 2013; Xiong, 2014) as an unstable wetting front is usually observed. The instability of the wetting front is accompanied by preferential flow. In this case, most of the water flows through the preferential pathways, while the rest of the porous medium remains dry even after hours of uniform infiltration. The preferential flow is also known as the finger-like flow because the main characteristic of this type of flow is the so-called fingers (DiCarlo, 2013). The finger consists of two parts, an undersaturated tail and an oversaturated tip, and this
non-monotonicity of saturation is called saturation overshoot. The physical description of preferential flow has been and still is a great challenge because it has huge application potential in the soil science (Lake, 1989; DiCarlo, 2013; Xiong, 2014) and in other fields (Bundt et al., 2000; Sutherland and Chase, 2008; Vafai, 2011).

Wetting front instability and associated saturation overshoot have been in the center of attention for several decades (Saffman and Taylor, 1958; Chuoke et al., 1959; Smith, 1967; Hill and Parlange, 1972). Since then, a huge number of laboratory and field



experimental work has become available. Some of the works concern 3D experiments (Glass et al., 1990; Yao and Hendrickx, 1996), but most of them are performed in 1D (long vertical tubes) (DiCarlo, 2004, 2007, 2010; Aminzadeh and DiCarlo, 2010) and in 2D (Hele-Shaw cells) (Smith, 1967; Glass et al., 1988, 1989b, c, a; Liu et al., 1994; DiCarlo et al., 1999; Glass et al., 2000; Bauters et al., 2000; Sililo and Tellam, 2005; Rezanezhad et al., 2006; Wei et al., 2014; Cremer et al., 2017; Pales et al., 2018; Chen et al., 2022; Liu et al., 2023) due to simpler realization. It turns out that flow in an unsaturated porous medium

has many unexpected features. For example, a non-monotonic dependence of the wetting front velocity and finger width on the initial saturation is observed (Bauters et al., 2000). At lower initial saturation, the wetting front is unstable with slow and wide fingers. With increasing initial saturation, the fingers first narrow and speed up and then slow and widen again until a stable wetting front is observed for a residual initial saturation. Another non-intuitive behavior is the dependence on applied influx (Glass et al., 1989c; Yao and Hendrickx, 1996; DiCarlo, 2013). The wetting front is stable at low infiltration fluxes, then

unstable within a certain range, and stable again at high infiltration fluxes.

Together with an unstable wetting front, the preferential nature of the flow, i.e., the heterogeneity of the water flow, is essential. Therefore, many attempts to quantify the heterogeneity of water flow in homogeneous soil were performed. Probably the first attempt was made by Bouma et al. (1978) using small infiltrometers. The proposed method was later used by Kneale and White (1984), who introduced the so-called bypassing ratio, which is the ratio of the preferential flow rate to the total

flow rate. Another attempt was made by Täumer et al. (2006) using the effective cross section. The authors applied this approach to measure preferential flow for field experiments. Furthermore, a degree of preferential flow was introduced in Lichner et al. (2011) to quantify the heterogeneity of water flow. The authors estimated the effective cross section and the degree of preferential flow from a saturation image of a vertical section of sandy soil.

The classical approach to model an unsaturated porous media flow is the Richards' equation (Richards, 1931), which is a

combination of the mass balance law and the Darcy-Buckingham law (Buckingham, 1907). The Richards' equation is diffusive in nature as it is unable to model a non-monotonic saturation profile in the case of uniform infiltration rate (Fürst et al., 2009). Therefore, many extensions of the Richards' equation known as continuum models have been proposed (Hassanizadeh et al., 2002; Eliassi and Glass, 2002; Brindt and Wallach, 2020; Cueto-Felgueroso et al., 2020; Beljadid et al., 2020; Roche et al., 2021; Ommi et al., 2022a, b). Other approaches are, for example, discrete (pore-scale) models (Lenormand et al.,

1988; Primkulov et al., 2018; Wei et al., 2022) and combination of discrete and continuum approaches (Glass and Yarrington, 1989, 2003; Liu et al., 2005; Liu, 2017, 2022; Liu et al., 2023).

One such combination is the model proposed by Vodák et al. (2022). The authors developed the semi-continuum model and its formal limit in the form of a partial differential equation with a Prandtl-type hysteresis operator (Visintin, 1993) under the derivative. It was shown that the semi-continuum model was able to correctly reproduce experiments of flow into a long vertical

tube (Kmec et al., 2019). In Kmec et al. (2021), the model was used to replicate the transition between unstable and stable wetting front for increasing initial saturation. Along with this, the model was shown to correctly capture the finger persistence and the flow across a heterogeneous porous medium. Finally, the strong non-monotonic dependence of the wetting front on the initial saturation for a point source infiltration was captured well (Kmec et al., 2023). However, there is still one essential thing missing: the dependence of 2D/3D preferential flow on applied flux in terms of finger width and finger spacing (Yao and





Hendrickx, 1996; Glass et al., 1989c; DiCarlo, 2013). In this paper, we want to demonstrate the ability of the model to correctly

capture the transition from stable to unstable flow for low infiltration fluxes (Yao and Hendrickx, 1996) and the transition from
unstable to stable flow for high infiltration fluxes (Glass et al., 1989c). This complicated transition is not yet captured by any
model. We also want to show that the model captures well the flow instability and finger width as a function of infiltration flux.
In addition, the relation between the saturation overshoot and the wetting front instability will be investigated.

## 2 Methods

In this section, we first introduce the governing equation, which is a formal limit of the semi-continuum model derived by
Vodák et al. (2022). Then, a proper discretization of the governing equation is presented together with the discretization of the
Prandtl-type hysteresis operator to provide a description of the semi-continuum model. The semi-continuum model is used to
describe the movement of the wetting liquid in a porous medium, specifically it is a multiphase flow model used for modeling
unsaturated porous media flow.

### 2.1 Governing equation

The governing equation is given by Eq. (1). It is a partial differential equation containing Prandtl-type hysteresis operator $P_H$
(Fig. 1) under the spatial derivative.

$$(K_{PS}\partial_t S - \partial_t P_H)(P_H - v) \geq 0, \quad \forall v \in [C_2, C_1], \ P_H \in [C_2, C_1]. \tag{1a}$$

$$\theta\partial_t S + \text{div}\left(\frac{\kappa}{\mu}\sqrt{k(S^-)}\sqrt{k(S^+)}\left((0,0,\rho g) - \nabla P_H\right)\right) = 0, \quad S^\pm(x_0, t) = \lim_{x \to x_0^\pm} S(x, t). \tag{1b}$$

In this equation, the porous medium is characterized by its porosity $\theta\,[-]$, intrinsic permeability $\kappa\,[\text{m}^2]$ and relative permeability
$k(S)\,[-]$. If the saturation is continuous, then $k(S) = \sqrt{k(S^-)}\sqrt{k(S^+)}$. The wetting phase (liquid) is characterized by its
saturation $S\,[-]$, density $\rho\,[\text{kgm}^{-3}]$, dynamic viscosity $\mu\,[\text{Pas}]$ and pressure $P\,[\text{Pa}]$. In a porous material that is not completely
filled with liquid, the pressure $P$ has the meaning of capillary pressure which is actually the tensile stress by which the liquid is
held in the pores. This pressure $P$ in the liquid phase is less than the pressure in the non-wetting phase (gas), which is assumed
to be zero, therefore $P$ becomes negative. The parameter $g\,[\text{ms}^{-2}]$ denotes acceleration due to gravity.

In soil physics, the relationship between the saturation $S$ and the pressure $P$ exhibits strong hysteresis and is known as
the retention curve. The retention curve has two main hysteresis branches: the wetting branch and the draining branch. Both
branches are similar in shape and are increasing functions of saturation and pressure. However, the shape of the retention curve
strongly depends on the size of the sample on which the measurement is performed (Larson and Morrow, 1981; Mishra and
Sharma, 1988; Zhou and Stenby, 1993; Perfect et al., 2004; Hunt et al., 2013; Ghanbarian et al., 2015; Silva et al., 2018).
Moreover, as the sample size decreases, the pore size variability within the sample also decreases and the retention curve
becomes flatter which was experimentally confirmed in Silva et al. (2018). In the case of an infinitesimal volume of porous
medium (i.e., a single pore), the main wetting and draining branches are constant and are referred to as the water entry and air





entry values (points). Let us note that the constant main branches are also obtained for a porous medium with zero pore size variability (Pražák et al., 1999). This is consistent with a flattening of the retention curve as the pore size variability decreases. The hysteretic relationship for an infinitesimal volume is defined in Eq. (1) by the Prandtl-type hysteresis operator $P_H$ (Visintin, 1993) and has the form shown in Fig. 1. The main wetting branch $C_1$ [Pa] (water entry value) and the main draining branch $C_2$ [Pa] (air entry value) are denoted by blue lines in Fig. 1.

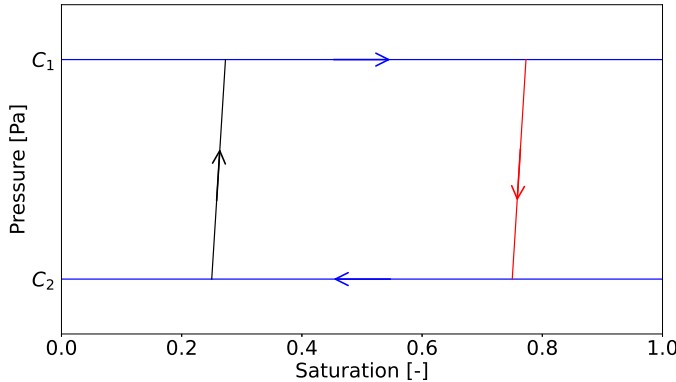

**Figure 1.** Prandtl-type hysteresis operator $P_H$ for an infinitesimal volume of porous medium. Blue lines represent constant main wetting branch $P(S) = C_1$ (water entry value) and constant main draining branch $P(S) = C_2$ (air entry value). Red and black lines are scanning curves with large gradient $K_{PS}$ – the direction of transition between the branches of the retention curve is indicated by the corresponding arrows.

There are many approaches to model the hysteresis between saturation and pressure (Mualem, 1976; Lenhard and Parker, 1987; Parker and Lenhard, 1987; Beliaev and Hassanizadeh, 2001; Abreu et al., 2019). Here, we use the following simple approach similar to the play-type hysteresis (Schweizer, 2017): If the saturation at a point in the material switches from wetting to draining, the pressure starts moving from the main wetting branch toward the main draining branch along the straight line with large gradient $K_{PS}$ (red line in Fig. 1). The same applies for the transition from the main draining branch to the main wetting branch (black line in Fig. 1). These transition curves are called scanning curves in soil physics. Note that the parameter $K_{PS}$ defined in Eq. (1) thus denotes the slope of the scanning curves.

The Prandtl-type hysteresis operator introduced in this way has the effect that the Eq. (1) switches between parabolic and hyperbolic types. If the pressure value is defined by the main branches, then the pressure-saturation relation is constant, and therefore $\nabla P_H = 0$. Thus, the equation becomes a hyperbolic differential equation. Otherwise the equation is a parabolic differential equation; the value of the pressure is given by the scanning curves in this case.

It is well known that different types of flow, saturated and unsaturated, can occur in parallel in the porous medium (Brand-horst et al., 2021). In a fully saturated medium, the pressure-saturation relation is no longer defined by the Prandtl-type hysteresis operator $P_H$. In this case, the pressure becomes hydrostatic pressure and takes on positive values. Using the hydrostatic pressure in Eq. (1) instead of $P_H$, we obtain the Laplace's equation as $k(S) = 1$ and $\partial_t S = 0$. Since in this paper we are focused on unsaturated flow, hydrostatic pressure is not implemented and the case of a fully saturated medium is not studied further.





## 2.2 Discretization of the porous medium

We want to simulate experiments in two dimensional Hele-Shaw cell of a porous medium, hence 2D discretization is used. The porous medium is a rectangle of size $A \times B$, where $A$ and $B$ denote the horizontal and vertical widths of the porous medium, respectively. The porous medium is represented by a square mesh consisting of $N \times M$ blocks (finite volumes) of size $\Delta x \times \Delta x$. These blocks retain the character of the porous medium.

## 2.3 Discretization of the Prandtl-type hysteresis operator

The discretization of the governing Eq. (1) has already been described in Kmec et al. (2023) as the semi-continuum model. The basic idea is to appropriately discretize the Prandtl-type hysteresis operator $P_H$ given by Eq. (1a). Its discretized version is the capillary pressure operator $P(S)$ which satisfies $P(S) \rightarrow P_H(S)$ for $\Delta x \rightarrow 0$. The basic idea of the discretization is that the shape of the retention curve depends on the size of the sample on which the measurement is performed (Silva et al., 2018). This fact is not ignored in our model; the sample size dependence is implemented so that the retention curve depends on the block size $\Delta x$. We refer to discretization as scaling of the retention curve. The proposed scaling is explained below, however for a detailed mathematical and physical justification we refer to the paper Vodák et al. (2022). For the reference block size $\Delta x_0$, the retention curve is given by the van Genuchten equation (Genuchten, 1980):

$$P_0^w(S) = -\frac{1}{\alpha_w}\left(S^{\frac{n_w}{1-n_w}} - 1\right)^{\frac{1}{n_w}}, \qquad\qquad P_0^d(S) = -\frac{1}{\alpha_d}\left(S^{\frac{n_d}{1-n_d}} - 1\right)^{\frac{1}{n_d}}, \qquad\qquad (2)$$

where $P_0^w$ is the main wetting branch, $P_0^d$ is the main draining branch, $\alpha_w, n_w$ are parameters of the main wetting branch, and $\alpha_d, n_d$ are parameters of the main draining branch. For a block size $\Delta x < \Delta x_0$, the main wetting and draining branches are scaled as follows:

$$P^w(S, \Delta x) = \frac{\Delta x}{\Delta x_0} P_0^w(S) + P_0^w(0.5)\left(1 - \frac{\Delta x}{\Delta x_0}\right), \qquad P^d(S, \Delta x) = \frac{\Delta x}{\Delta x_0} P_0^d(S) + P_0^d(0.5)\left(1 - \frac{\Delta x}{\Delta x_0}\right). \qquad (3)$$

Obviously, for $\Delta x = \Delta x_0$, the retention curve is given by Eq. (2). For $\Delta x \rightarrow 0$, the retention curve converges to the Prandtl-type hysteresis operator $P_H$ so that $C_1 = P_0^w(0.5)$ and $C_2 = P_0^d(0.5)$. Hence, $C_1$ and $C_2$ represent the constant limits of the main wetting and draining branches, respectively. Note that the reference block size $\Delta x_0$ is a parameter of the semi-continuum model that is not arbitrary. For instance, the parameter $\Delta x_0$ was calibrated for 20/30 sand in Kmec et al. (2023) using the experiments of Bauters et al. (2000). Figure 2 shows the capillary pressure operator $P(S)$ for different block sizes. It can be clearly seen that for $\Delta x \rightarrow 0$ the operator $P(S)$ converges to the Prandtl-type hysteresis operator $P_H$ shown in Fig. 1.

Although it is well known that the retention curve is dependent on the sample size of the porous medium (Larson and Morrow, 1981; Mishra and Sharma, 1988; Zhou and Stenby, 1993; Perfect et al., 2004; Hunt et al., 2013; Ghanbarian et al., 2015), the implementation of this dependence is not common in flow modeling. Moreover, other characteristics of the porous medium, such as permeability and porosity, are also dependent on the sample size (Mishra and Sharma, 1988; Ewing et al., 2010; Ghanbarian et al., 2017, 2021; Esmaeilpour et al., 2021). In the semi-continuum model, the sample size dependence of





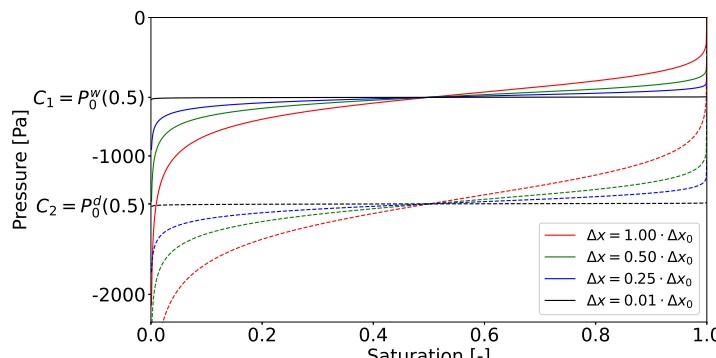

**Figure 2.** The scaling of the retention curve for different block sizes $\Delta x$. The solid lines denote the main wetting branches and the dashed lines denote the main draining branches. The parameters $\alpha_w, n_w, \alpha_d, n_d$ are given in Table 1. For $\Delta x \to 0$, the retention curve converges to the Prandtl-type hysteresis operator $P_H$ such that the main wetting and draining branches rotate around values $C_1$ and $C_2$.

the retention curve is implemented by a proper scaling of the retention curve. In principal, it means that individual blocks of the discretization mesh represent a real sample of the porous medium, hence these blocks carry the information about the physical characteristics. This differs fundamentally from standard numerical schemes for partial differential equations, where the mesh plays only a mathematical role and ignores the fact that the individual elements/volumes represent the real domain. However, some authors have already taken this fact into account in modeling porous media. For instance, White et al. (2006) estimated a lower limit of finite elements and then used this size in their model. They argue that the use of smaller elements would not be appropriate because it would lead to violation of the continuum assumptions.

### 2.4 Discretization of the governing equation – the semi-continuum model

Each block of the discretized porous medium is denoted by the indices $(i,j)$ representing the corresponding row and column. $S_t(i,j)\,[-]$ and $P_t(i,j)\,[\mathrm{Pa}]$ denote the saturation and the pressure of the wetting phase (liquid) of the block $(i,j)$ at time $t$, respectively. The both values are assumed to be constant within the block. Moreover, $q_{t\,(i_2,j_2)}^{(i_1,j_1)}\,[\mathrm{ms}^{-1}]$ denotes the flux of the wetting phase from the block $(i_1,j_1)$ to the block $(i_2,j_2)$ at time $t$.

The semi-continuum model, i.e., the discretization of the governing Eq. (1), consists of three consecutive steps: saturation update, pressure update and flux update. First, the saturation in each block is updated according to discretized mass balance law:

$$S_{t+\Delta t}(i,j) = S_t(i,j) + \frac{\Delta t}{\theta}\frac{1}{\Delta x}\left(q_{t\,(i,j)}^{(i-1,j)} - q_{t\,(i+1,j)}^{(i,j)} + q_{t\,(i,j)}^{(i,j-1)} - q_{t\,(i,j+1)}^{(i,j)}\right), \tag{4}$$

where $\Delta t$ is a time step and $\Delta x$ is the block size.

The second step is to update the pressure in each block to obtain the pressure at time $t+\Delta t$, i.e., $P_{t+\Delta t}$. The pressure is updated according to the capillary pressure operator $P(S)$ which is a discretized Prandtl-type hysteresis operator. The operator $P(S)$ consists of the main wetting and draining branches given by Eq. (3). For a complete definition of the capillary pressure





operator it is necessary to include the hysteresis. All the scanning curves between the two main branches are straight lines with large gradient $K_{PS}$ (see red and black lines in Fig. 1).

The third and final step is the flux update. Let us define the effective permeability as $\gamma(S) = \kappa k(S)$. For the relative permeability function $k(S)$ we use the form derived in Genuchten (1980):

$$k(S) = S^\lambda \left[ 1 - \left( 1 - S^{\frac{n}{n-1}} \right)^{\frac{n-1}{n}} \right]^2,$$ (5)

where $\lambda\,[-]$ is a free parameter and $n\,[-]$ is a parameter of the retention curve given by Eq. (2). The flux between blocks is updated using the discretized version of Darcy-Buckingham law (Bear, 1972):

$$q_{t+\Delta t (i_2,j_2)}^{(i_1,j_1)} = \begin{cases} \frac{1}{\mu} \sqrt{\gamma(S_{t+\Delta t}(i_1,j_1))\gamma(S_{t+\Delta t}(i_2,j_2))} \left( \rho g - \frac{P_{t+\Delta t}(i_2,j_2) - P_{t+\Delta t}(i_1,j_1)}{\Delta x} \right), & \text{for } j_1 = j_2,\ i_2 = i_1 + 1 \\ \frac{1}{\mu} \sqrt{\gamma(S_{t+\Delta t}(i_1,j_1))\gamma(S_{t+\Delta t}(i_2,j_2))} \left( 0 - \frac{P_{t+\Delta t}(i_2,j_2) - P_{t+\Delta t}(i_1,j_1)}{\Delta x} \right), & \text{for } i_1 = i_2,\ j_2 = j_1 + 1 \\ 0, & \text{otherwise} \end{cases}$$ (6)

The acceleration due to gravity is included only for the vertical fluxes. Unsurprisingly, the fluxes between non-neighboring blocks are set to zero. Note that the geometric mean is used to average the effective permeability between blocks which is consistent with the governing Eq. (1). Moreover, this type of averaging is also consistent with the experimental observation (Jang et al., 2011). After updating the fluxes between neighboring blocks, we update the time $t = t + \Delta t$ and return to the saturation update given by Eq. (4).

If the fluxes between blocks are too large, especially if the flux is close to the saturated conductivity $K_S = \frac{\kappa}{\mu}\rho g$, the saturation may exceed one. This is often the case for other models as well. For example, in Cueto-Felgueroso and Juanes (2009), a "compressibility term" is used for the capillary energy-saturation dependence. This term becomes dominant near saturation close to one, so it prevents the saturation to increase any further. We use a different approach; the magnitude of the flux to the block can be at most so large that the saturation does not exceed one. This straightforward approach is only possible because of the simple numerical scheme used.

Let us note that various discretizations of the Prandtl-type hysteresis operator $P_H$ can be proposed. However, the used linear scaling of the retention curve is physically justifiable. It is convenient to preserve the same fluxes between blocks for various block sizes. According to Eq. (6), decreasing the block size $\Delta x$ by half doubles the flux. Therefore, the linear scaling is introduced in Eq. (3) so that the fluxes remain the same as the block size decreases.

According to Eq. (4) and Eq. (6), if a standard retention curve without scaling is used, i.e., it does not converge to a Prandtl-type hysteresis operator $P_H$, the semi-continuum model degenerates into a numerical scheme for solving the classical Richards' equation. Unlike the semi-continuum model, the Richards' equation is unable to admit finger-like solutions regardless of the hysteresis used (Fürst et al., 2009). From a mathematical point of view, in the case of an unsaturated porous medium, the Richards' equation is only a parabolic differential equation compared to the governing Eq. (1), which is a hyperbolic-parabolic differential equation.



## 3 Results

### 3.1 Experimental setup

We want to reproduce the dependence of the wetting front stability on different infiltration rates (Glass et al., 1989c; Yao and Hendrickx, 1996; DiCarlo, 2013). We first briefly describe the experiments of Glass et al. (1989c) and Yao and Hendrickx (1996). Although these experiments are approximately 30 years old, they have not yet been fully reproduced by any model. Glass et al. (1989c) infiltrated into a two-dimensional chamber with a thickness of $1\,\mathrm{cm}$. Water was uniformly applied at a constant flux $q_{\mathrm{top}}$ at the top boundary of the porous medium. They showed that the finger width increases with increasing applied flux until a stable wetting front is observed when the flux is close to the saturated conductivity $K_S$. Yao and Hendrickx (1996) performed similar experiments, but they infiltrated into large three-dimensional columns (with diameters $30\,\mathrm{cm}$ and $100\,\mathrm{cm}$) and the applied flux was much lower, approximately between $0.001 - 0.45\,\mathrm{cm\,min}^{-1}$. They demonstrated that as the flux decreases, the finger-like flow disappears and a stable wetting front reappears. Therefore, the finger-like flow is observed only within a certain range of applied flux. This is not consistent with stability arguments of Chuoke et al. (1959) and Parlange and Hill (1976) as they do not predict an increase in finger width towards very low fluxes.

Moreover, Yao and Hendrickx (1996) used four different sands (14/20, 20/30, 30/40 and 40/60) varying in the size of particles from coarser to finer. The authors showed that the transition from an unstable to a stable flow for low fluxes is maintained regardless of the used sand. They also demonstrated that fingers tend to be wider for finer sand compared to coarser sand. This is expected because a stable wetting front is more readily observed for finer sand (Cremer et al., 2017). In this paper, we decided to use only one type of sand, since it has already been shown in Kmec et al. (2023) that the semi-continuum model captures this dependence well and that the fingers are indeed wider for finer sands (for details we refer to section B3 in Kmec et al. (2023)).

In Kmec et al. (2023) we used the semi-continuum model to correctly reproduce the experiments reported in Bauters et al. (2000) – the non-monotonic dependence of the saturation overshoot and the finger velocity on the initial saturation, the so-called Bauters' paradox. Here we use the same parameters as in Kmec et al. (2023), including $\Delta x_0$, to avoid the possibility that we have adjusted the parameters of the semi-continuum model to obtain the best results. Thus, without additional parameter adjustment, we want to reproduce completely different phenomena of porous media flow. Parameters used for reproducing the dependence on different infiltration rates are given in Table 1. The porous medium used for simulations is 20/30 sand. Let us note that the value of $K_{PS}$ does not affect the results if it is chosen large enough. The differences between solutions are negligible for $K_{PS} \geq 10^5\,\mathrm{Pa}$ (see Fig. 3.13 in Kmec (2021)). Here we use the lower limit $K_{PS}$.





**Table 1.** Parameters used for reproducing the dependence on different infiltration rates. Parameters for 20/30 sand were adopted from Schroth et al. (1996) and DiCarlo (2004).

| Parameter | Symbol | Value |
|---|---|---|
| Horizontal width of the chamber | $A$ | 50 cm |
| Vertical width of the chamber | $B$ | 50 cm |
| Reference block size | $\Delta x_0$ | $\frac{10}{12}$ cm |
| Block size | $\Delta x$ | 0.25 cm |
| Porosity | $\theta$ | 0.35 |
| Density of water | $\rho$ | 1000 kgm$^{-3}$ |
| Dynamic viscosity of water | $\mu$ | $9 \times 10^{-4}$ Pas |
| Intrinsic permeability | $\kappa$ | $2.294 \times 10^{-10}$ m$^2$ |
| Relative permeability exponent | $\lambda$ | 0.8 |
| Acceleration due to gravity | $g$ | 9.81 ms$^{-2}$ |
| Wetting curve parameter | $\alpha_w$ | 0.177 cm$^{-1}$ |
| Wetting curve parameter | $n_w$ | 6.23 |
| Draining curve parameter | $\alpha_d$ | 0.0744 cm$^{-1}$ |
| Draining curve parameter | $n_d$ | 8.47 |
| Slope of scanning curves | $K_{PS}$ | $10^5$ Pa |
| Initial saturation | $S_{in}$ | 0.01 |
| Residual saturation | $S_{rs}$ | 0.05 |

The scheme of the experimental setup is shown in left panel of Fig. 3. The porous medium is initially dry (initial saturation $S_{in} = 0.01$), and all the blocks begin on the main wetting branch. Boundary conditions are set to be consistent with the experiments we want to reproduce (Yao and Hendrickx, 1996; Glass et al., 1989c). The constant infiltration rate $q_{top}$ is applied at the whole top boundary. In total, we used 18 different infiltration rates $q_{top}$, with the lowest influx equal to $0.001 \, \mathrm{cm \, min^{-1}}$ and the highest influx equal to the saturated conductivity $K_S = 15 \, \mathrm{cm \, min^{-1}}$. The lateral boundaries of the porous medium are impenetrable, so the lateral fluxes are set to zero. For the bottom boundary flux $q_{bot}$, a free discharge is prescribed:

$$q_{bot} := q_{t(out)}^{(N,j)} = \begin{cases} 0 & \text{for } S_t \leq S_{rs} \\ \frac{1}{\mu} \gamma(S_t(N,j)) \left( \rho g + \frac{P_t(N,j)}{\Delta x} \right), & j = 1, \ldots, M, \quad \text{for } S_t > S_{rs} \end{cases}, \tag{7}$$

where $N$ denotes the bottom row index. Thus, the flux from the bottom boundary is set to zero if the saturation of the corresponding block does not exceed the residual saturation $S_{rs}$, otherwise it is non-zero. This implementation is standard for the models based on the Richards' equation (Šimůnek and Suarez, 1994) and for the semi-continuum model was already used in Kmec et al. (2021). Moreover, if the porous medium is homogeneous, this does not mean that all characteristics, such as the intrinsic permeability, are exactly the same for each block. Therefore, a small distribution of spatially correlated intrinsic



permeability is included (see right panel of Fig. 3). A similar distribution has been also used for example in Cueto-Felgueroso and Juanes (2009); Gomez et al. (2013); Kmec et al. (2023).

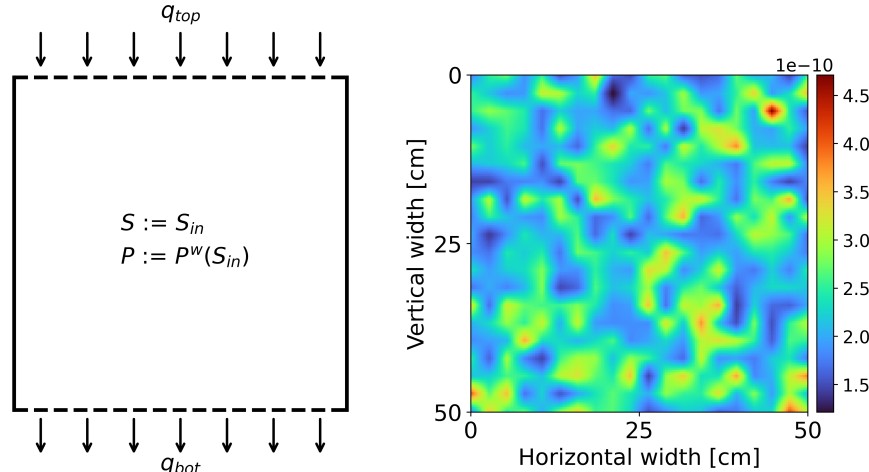

**Figure 3.** Left panel: The scheme of the experimental setup. Right panel: The distribution of spatially correlated intrinsic permeability. The average value of $\kappa$ equals approximately $2.294 \times 10^{-10}\,\mathrm{m}^2$ and the distribution satisfies $\kappa_{\max}/\kappa_{\min} \approx 4$. The values are colored according to the color bar on the right.

### 3.2   Evolution of the saturation profile

Figure 4 shows the evolution of the saturation profile at six different times for $q_{\mathrm{top}} = 0.05\,\mathrm{cm}\,\mathrm{min}^{-1}$. Times are displayed in the upper left corner for each frame. First, a stable wetting front with small frontal perturbations is developed for time

$t = 300\,\mathrm{s}$. These perturbations then grow into long persistent fingers. Finally, when fingers reach the bottom of the chamber, the water flows out of the chamber through preferential pathways so that the most of the porous medium remains dry. This evolution of the wetting front instability is consistent with the experimental observation (DiCarlo, 2013). Note that if small frontal perturbations do not grow, stable flow is produced.

### 3.3   Dependence of flow on infiltration rate

Let us now examine the dependence of flow on different infiltration rates. In total, 18 simulations for infiltration rates in range $q_{\mathrm{top}} = 0.001 - 15\,\mathrm{cm}\,\mathrm{min}^{-1}$ are performed. Saturation profiles for nine simulations are shown in Fig. 5. For the sake of clarity, saturation profiles of all simulations are given in Appendix A1; see Figs. A1 and A2. The time for each flux is chosen so that the saturation reaches $40\,\mathrm{cm}$ from the upper boundary. In addition, the flux and the corresponding time are displayed in the upper left corner for each frame. At low fluxes, the transition from a stable wetting front to the finger-like flow is well observed.

Moreover, if the flux is close to the hydraulic conductivity, the fingers widen and a stable wetting front develops. This is the



first model that is able to simulate this non-trivial transition. A more detailed view can be found in the videos, where transient simulations can be seen. The videos are available in Kmec (2023b) for each applied influx.

It is known that in unstable flow, two fingers can merge or one finger can split into two fingers (Glass et al., 1989b, c; Rezanezhad et al., 2006). Both scenarios are reproduced here, but to see details, we recommend to look at videos of the

transient simulations that can be downloaded from Kmec (2023b). The merging can be seen, e.g., for $q_{\text{top}} = 5 \, \text{cm} \, \text{min}^{-1}$, where two wide fingers are merged. The finger merging is also well observed for lower fluxes. Furthermore, the splitting of the finger into two can be observed for $q_{\text{top}} = 2.5 \, \text{cm} \, \text{min}^{-1}$.

Note that the saturation overshoot is clearly evident for fluxes between $0.05 - 5 \, \text{cm} \, \text{min}^{-1}$. Moreover, even for lower fluxes, for which an unstable behavior is still present, the saturation overshoot can be observed. However, the magnitude of the

saturation overshoot (i.e., the saturation difference between finger tip and tail) is very small, so that it is not visible in Fig. 5. This is discussed in more detail in Sect. 3.6 *Wetting front instability*.

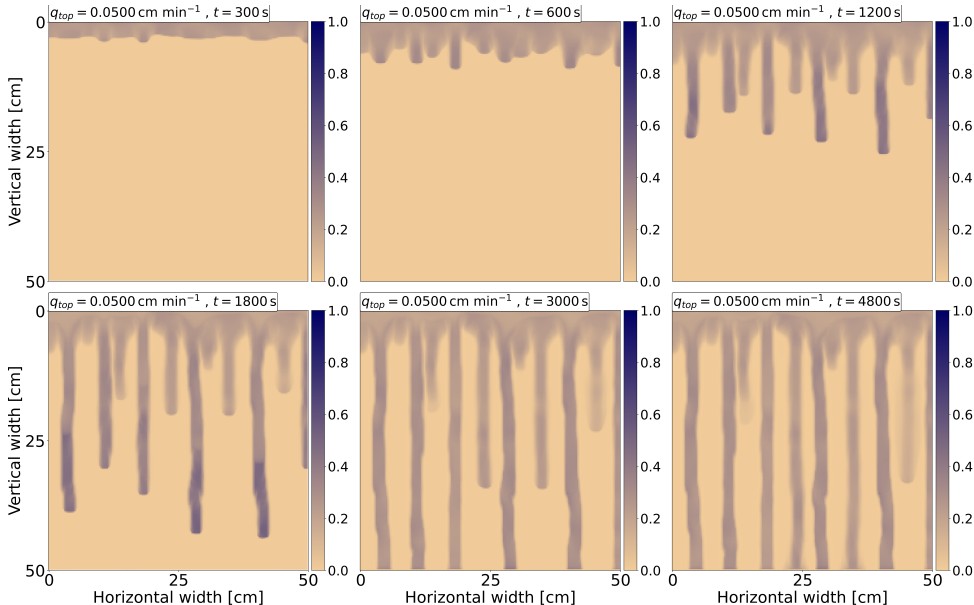

**Figure 4.** Evolution of saturation profile for $q_{\text{top}} = 0.05 \, \text{cm} \, \text{min}^{-1}$ at six different times. Times are displayed in the upper left corner for each frame. Saturation is colored according to the color bar on the right.





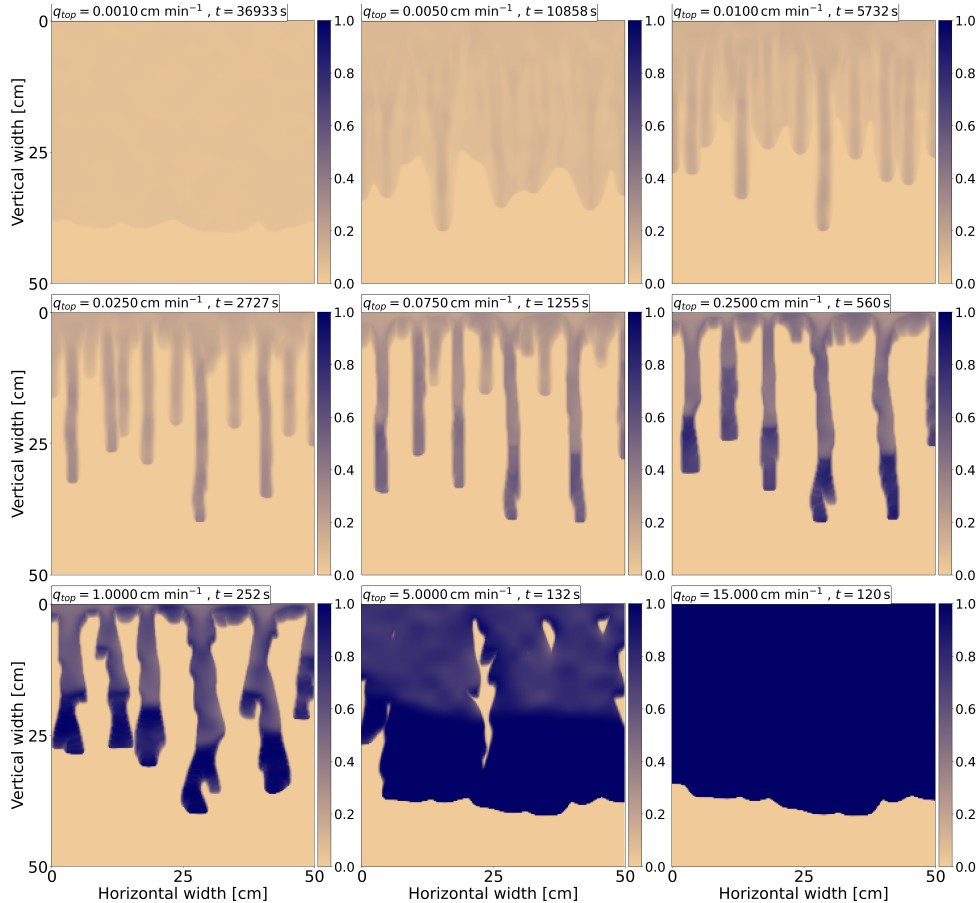

**Figure 5.** Saturation profiles for nine different infiltration rates. For each frame, the influx is displayed together with the simulation time. The transition from stable to unstable and back to stable flow is well observed. Saturation is colored according to the color bar on the right.

### 3.4 Finger width as function of influx

DiCarlo (2013) plotted the measured finger widths as a function of influx in a single graph together for low (Yao and Hendrickx, 1996) and high (Glass et al., 1989c) infiltration rates. The measured finger widths for both experiments are shown in Figure 2 in DiCarlo (2013) along with the predicted finger widths using standard theory (Chuoke et al., 1959; Parlange and Hill, 1976). The observed results can be summarized as follows:

- A stable wetting front is observed at very low fluxes – in this case the finger width is equal to the chamber width. This is not predicted by standard theory (Chuoke et al., 1959; Parlange and Hill, 1976).

- As the influx increases, a rapid decrease in the finger width is observed followed by a long flat valley of almost constant finger width for different fluxes – specifically, the finger width first slightly decrease and then slightly increase.



- Finally, if the magnitude of the flux is getting close to the saturated conductivity, the finger widths increase again, followed by stable flow.

Let us now calculate the finger width on the influx for all performed simulations. For each influx, we use the saturation profile at the time of the simulation for which the water reaches the bottom of the chamber, i.e., $50\,\mathrm{cm}$. Therefore, the bottom boundary flux given by the Eq. (7) does not influence the saturation profile. However, in the case of unstable flow, the calculated finger widths are the same if a much longer simulation time is used, so that water flows out of the porous medium. This is due to the fact that the fingers are persistent in time and do not disappear even after several hours of steady infiltration (Glass et al., 1989b; Rezanezhad et al., 2006). Since this finger persistence is captured well by the semi-continuum model (Kmec et al., 2021) (see also Fig. 4), the simulation time used for finger width calculation in the case of unstable flow is irrelevant. To calculate the finger width, a segmentation of all fingers is done for each influx. In the case of finger-like flow, this segmentation is straightforward because fingers are well developed. For a stable flow, the entire saturation profile is assumed to be a "finger" (the same notation is used in DiCarlo (2013)). However, a more complex case is the intermediate phase between stable and unstable flow (see $q_{\mathrm{top}} = 0.01\,\mathrm{cm\,min}^{-1}$ in Fig. 5) – the transition between unstable and stable flow. Some fingers are narrow and some are diffusely expanding, so a completely objective segmentation of the fingers is not possible here. For clarity, all performed finger segmentations are included in Appendix A2 (see Fig. A3). Note that fingers in contact with the edge of the chamber are not included in the analysis, as they were also not included in Glass et al. (1989c). These "side" fingers tend to be slightly narrower (see Fig. 5), which is consistent with the experimental observations (Glass et al., 1989c).

Figure 6 shows the dependence of the calculated finger width on the influx. Red, green and blue dots indicate fluxes for which we observe stable flow, a transition between stable and unstable flow, and unstable flow, respectively. The results are consistent with experimentally observed behavior – a nearly constant finger width is observed for flows between $0.01 - 2.5\,\mathrm{cm\,min}^{-1}$, followed by a stable flow for very low and very high applied fluxes. Even a slight increase in the finger width for fluxes above $0.01\,\mathrm{cm\,min}^{-1}$ is in perfect agreement with experiments (see Figure 2 in Glass et al. (1989c)). The specific case is $q_{\mathrm{top}} = 5\,\mathrm{cm\,min}^{-1}$ (see Fig. 5); two fingers are first developed and then both fingers merge at a depth of approximately $30\,\mathrm{cm}$. The average width of the two fingers or of the one merged finger can be calculated. In Fig. 6 the second case is selected, i.e., the width is calculated from one merged finger.

Glass et al. (1989c) calculated the number of fingers including side fingers for each applied flux. They reported that in the case of unstable flow, the number of fingers does not change significantly for different fluxes, as it varies between four and six. Given the size of the chamber used for the experiments ($30\,\mathrm{cm}$), the expected number of fingers for the $50\,\mathrm{cm}$ used in our simulations is approximately between seven and ten. This corresponds to the number of fingers developed in the simulations, which for unstable flow range from six to ten. Hence, the spacing between the fingers is also captured well in the simulations.





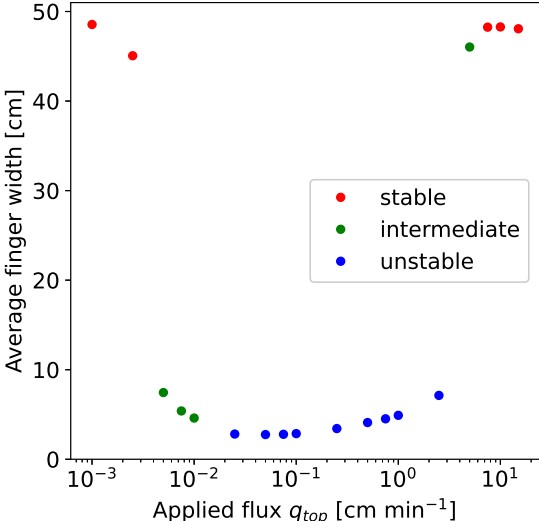

**Figure 6.** Dependence of the finger width on the influx. A stable wetting front is observed at either very low or very high applied fluxes – in this case the "finger width" is approximately equal to the horizontal width of the chamber. The color of dots indicate the type of flow: stable, intermediate and unstable flow.

### 3.5 Preferential flow as function of influx

Let us now examine how much the flow is preferential depending on the influx. One possible approach is to calculate for a given saturation profile the fraction of wetted area to the total area of a porous medium of size $50\,\text{cm} \times 50\,\text{cm}$. For each top boundary flux, the saturation profile is chosen at the simulation time when the flow is already stabilized. This means that analysis of the saturation profiles over a longer simulation period leads to the same conclusions, because the saturation profile does not change even if the inflow continues. Figure 7 shows the fraction of wetted area versus influx. It is not surprising that in the case of unstable flow the ratio of wetted area to the total area of a porous medium is mostly between $40 - 60\,\%$, while for stable flow it is $100\,\%$. Note that the porous medium is completely wetted for the intermediate cases as well. However, the fraction of wetted area is not an appropriate metric for calculating how much water flows preferentially. This is because this fraction says nothing about whether the majority of the water flows through only a small part of the porous medium. Therefore, a by-pass ratio in the horizontal section based on a similar approach to that of Kneale and White (1984) is used instead.

The by-pass ratio is calculated as follows: First, the inflow to the block is calculated for each block that corresponds to the horizontal section at a depth of $30\,\text{cm}$. The total number of blocks is $200$ as the horizontal width of the porous medium is $50\,\text{cm}$ and the block size $\Delta x = 0.25\,\text{cm}$. The inflows to the blocks are then normalized so that the average inflow of one block is equal to one. Using this type of normalization, the inflow values for perfectly stable flow are equal to one for all blocks. These values represent the by-pass ratio. The simulation times are the same as for calculation the fraction of wetted area to the total area shown in Fig. 7. Therefore, the analysis in longer simulation times does not affect the obtained results.



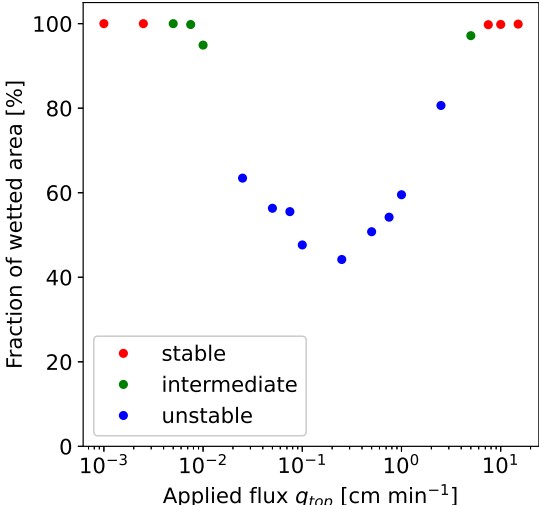

**Figure 7.** Ratio of wetted area to the total area of a porous medium of size $50\,\text{cm} \times 50\,\text{cm}$ depending on influx. For each influx, the saturation profile is chosen at a sufficiently long simulation time so that the flow is already stabilized. The color of dots indicate the type of flow: stable, intermediate and unstable flow.

Left panel of Fig. 8 shows the by-pass ratio in the horizontal section at a depth of $30\,\text{cm}$ for three different influxes. The color indicates the type of flow: red, green and blue colors denote an example of stable, intermediate and unstable flow, respectively.

The by-pass ratio for stable flow is equal to one almost everywhere with no significant preference of flow. For unstable flow, the by-pass ratio corresponds to the developed fingers. Water flows only through the fingers and outside the finger the flow is zero. Quite surprising is the intermediate case – the transition between stable and unstable flow. Water flows through the entire porous medium, but the flow is still highly preferential. This is counterintuitive since the porous medium is fully wetted in this case, yet preferential pathways are formed through which the most of the water flows. The origin of these pathways can be

seen in Fig. 5 for $q_{\text{top}} = 0.005\,\text{cm}\,\text{min}^{-1}$, where they appear as a slight increase in saturation. However, for better illustration, right panel of Fig. 8 shows the saturation profile at a longer time ($t = 24000\,\text{s}$). Moreover, the maximum value of the color bar is changed to make pathways more visible. The created pathways are well observed and they do not disappear even when the porous medium is completely wetted. Comparing the by-pass ratio for $q_{\text{top}} = 0.005\,\text{cm}\,\text{min}^{-1}$ (left panel of Fig. 8) and the corresponding saturation profile (right panel of Fig. 8), we can clearly see that the highest by-pass ratio corresponds to the

most saturated parts of the porous medium. Note that the similar scenario holds for the other two intermediate cases for fluxes $0.0075$ and $0.01\,\text{cm}\,\text{min}^{-1}$, where the resulting pathways are more pronounced – for details, see the plot of the by-pass ratio for all the applied fluxes in Fig. A4 in Appendix A3. We conjecture that this is a very important observation about the problem of preferential flow in unsaturated porous media, since experimental measurements are highly limited for this case. Using the semi-continuum model, more details can be observed and therefore it is easier to understand the origin of the preferential

pathways for different boundary conditions.





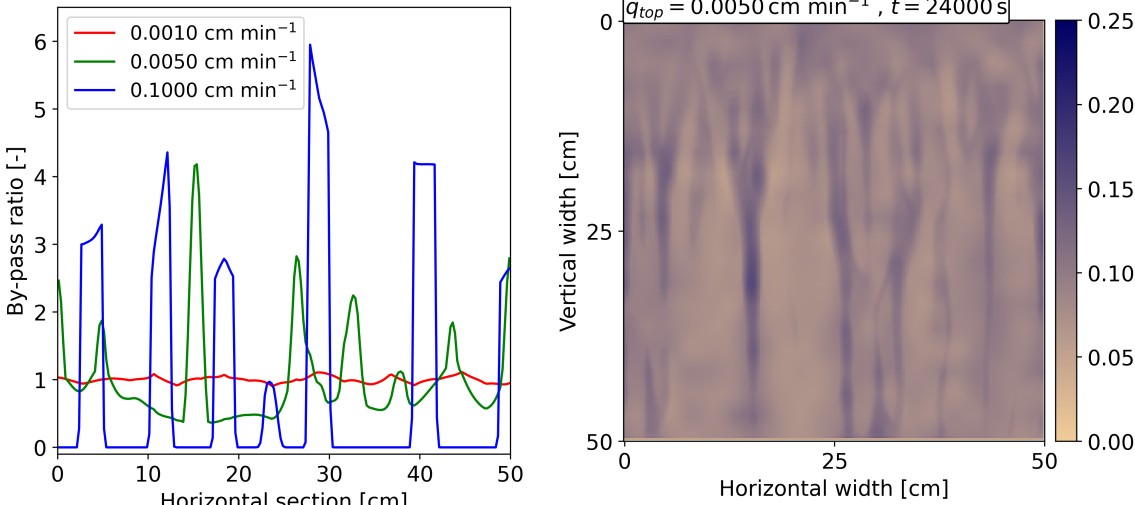

**Figure 8.** Left panel: The by-pass ratio for stable (red), intermediate (green) and unstable flow (blue) at a depth of $30\,\mathrm{cm}$ of the porous medium. Stable and unstable flows behave as expected. For intermediate case, water still flows preferentially when the porous medium is fully wetted. Right panel: Saturation profile for $q_{\mathrm{top}} = 0.005\,\mathrm{cm\,min}^{-1}$ at time $t = 24000\,\mathrm{s}$. The simulation corresponds to the intermediate case in left panel of this figure. Saturation is colored according to the color bar on the right. The pathways of the saturation are clearly visible in the image and correspond to the highest by-pass ratio values.

It is also useful to represent the preferential flow by a single value. This is done by calculating the smallest number of blocks through which at least $50\,\%$ of the total amount of water flows. The same horizontal section as for calculating the by-pass ratio is used, i.e., at a depth of $30\,\mathrm{cm}$. In principal, the inflows to the blocks are sorted from the largest value and then it is determined for how many blocks is the sum of the inflows at least $50\,\%$ of the sum of all the inflows to the blocks. Moreover, for each influx, the length is calculated as $L = \Delta x \times n_B$, where $n_B$ denotes the calculated number of blocks. In the case of perfectly stable flow, the length is equal to half the horizontal width of the porous medium, i.e., $25\,\mathrm{cm}$. Figure 9 shows the calculated length depending on influx. In the case of stable flow, half of the total amount of water indeed flows through almost $25\,\mathrm{cm}$. On the other hand, the preferential flow is dominant for unstable flow, where the length is between $5.50 - 8.75\,\mathrm{cm}$. The exception is $q_{\mathrm{top}} = 2.5\,\mathrm{cm\,min}^{-1}$, for which it is $12.50\,\mathrm{cm}$ because the fingers are significantly wider compared to the lower fluxes. Finally, for fluxes $q_{\mathrm{top}} = 0.005 - 0.01\,\mathrm{cm\,min}^{-1}$ (intermediate case), the length is similar to values of unstable flow, and significant preferential flow is still observed. This is consistent with the by-pass ratio analysis.





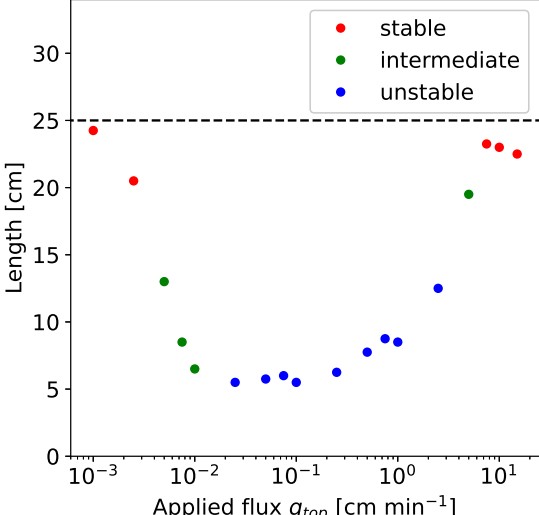

**Figure 9.** The length of the porous medium in the horizontal section at a depth of $30\,\mathrm{cm}$ through which $50\,\%$ of the water flows is plotted against the influx. For each influx, the simulation times are taken so that the flow through the porous medium does not change any further. The dashed line denotes the case of perfectly stable flow, i.e., $25\,\mathrm{cm}$. The color of dots indicate the type of flow: stable, intermediate and unstable flow.

### 3.6 Wetting front instability

There is mathematical and experimental evidence that the saturation overshoot in 1D is closely related to the wetting front instability (Eliassi and Glass, 2001; Egorov et al., 2003; DiCarlo, 2013). In other words, the cause of the wetting front instability

is the saturation and pressure overshoot (Eliassi and Glass, 2001). For instance, it was shown experimentally that the flux range for the saturation overshoot in 1D (DiCarlo, 2004) corresponds to the flux range for the 3D unstable flow (Yao and Hendrickx, 1996; Glass et al., 1989c). The same conclusion was obtained for the dependence on initial saturation when comparing 1D and 2D experiments (DiCarlo, 2004; Bauters et al., 2000). In summary, if no overshoot is observed in 1D, the flow is assumed to be stable in 2D/3D (Raats, 1973; Egorov et al., 2003; van Duijn et al., 2004). On the other hand, if the saturation overshoot is

observed in 1D, the 2D/3D flow is predicted to be unstable (Egorov et al., 2003; Nieber et al., 2005). This simplifies the analysis of the wetting front instability because we can switch to 1D where the analysis is easier. Moreover, according to the stability analysis of Saffman and Taylor (1958), the flow is predicted to be stable if the influx is larger than the saturated conductivity, i.e., $q_{\mathrm{top}} \geq K_S$. Otherwise, the flow is predicted to be unstable. For fluxes above the saturated conductivity, the condition is indeed valid and the flow is stable (DiCarlo, 2013). On the contrary, the condition is not valid for lower fluxes as the flow is

observed to be stable for very low fluxes (Yao and Hendrickx, 1996).

     In this section, we want to analyze the consistency between the saturation overshoot in 1D and the instability of the wetting front in 2D. The 1D simulations are performed with the same parameters as 2D simulations (see Table 1) with two exceptions: the horizontal width of the chamber $A$ is equal to the size of the block $\Delta x$, i.e., $A = 0.25\,\mathrm{cm}$ and the distribution of the intrinsic



permeability is not used. Again, 18 simulations are conducted for infiltration rates in range $q_{\text{top}} = 0.001 - 15.00 \,\text{cm}\,\text{min}^{-1}$.

Left panel of Fig. 10 shows saturation profiles for six different applied fluxes $q_{\text{top}}$. For the lowest influx $q_{\text{top}} = 0.001 \,\text{cm}\,\text{min}^{-1}$, the profile is stable without saturation overshoot. For $q_{\text{top}} = 0.010 \,\text{cm}\,\text{min}^{-1}$, the saturation overshoot is formed and is more pronounced with increasing influx up to $q_{\text{top}} = 1.000 \,\text{cm}\,\text{min}^{-1}$. Then, the saturation overshoot becomes less pronounced, until it disappears completely and the stable profile is observed again for the highest influx $q_{\text{top}} = 15.00 \,\text{cm}\,\text{min}^{-1}$. This is in a good agreement with 1D experiments (see figure 2 in DiCarlo (2010)). Note that the occurrence of the saturation overshoot

is consistent with 2D simulations.

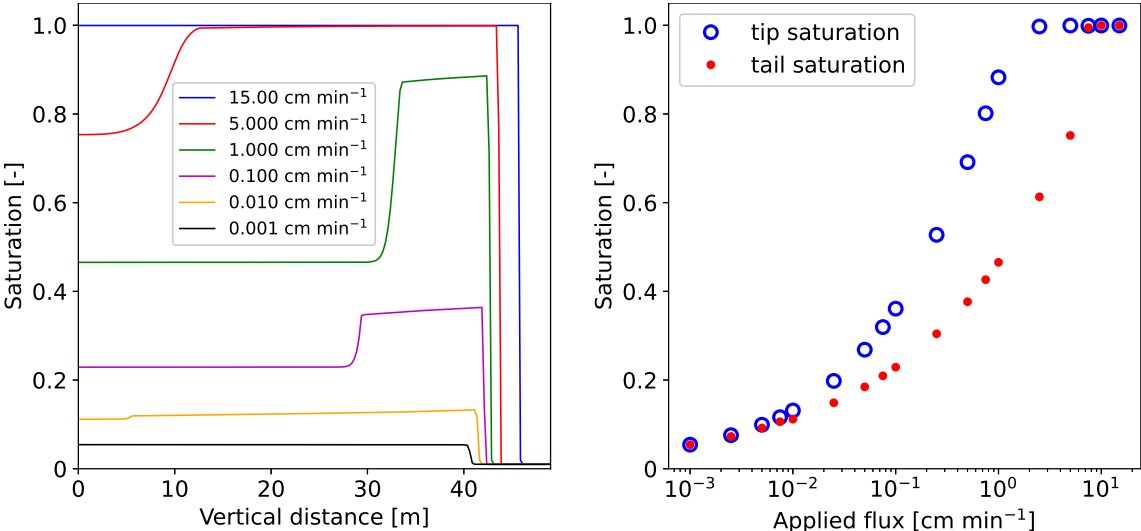

**Figure 10.** Left panel: Saturation profiles for various applied fluxes. The saturation overshoot is observed for intermediate fluxes, but not for the lowest flux ($q_{\text{top}} = 0.001 \,\text{cm}\,\text{min}^{-1}$) and the highest flux ($q_{\text{top}} = 15.00 \,\text{cm}\,\text{min}^{-1}$). Right panel: Tip and tail saturation dependence on the applied flux. The occurrence of the saturation overshoot is consistent with the instability of 2D flow.

Right panel of Fig. 10 shows the saturation of finger tip and tail for 18 different infiltration rates. The difference between the tip and tail represents the magnitude of the saturation overshoot. The obtained results are again in agreement with 1D experiments (see figure 3 in DiCarlo (2010)). We can see a perfect consistency between 1D saturation overshoot and 2D wetting front. Specifically, for applied fluxes for which a stable wetting front is observed in 2D, no overshoot is observed in

1D. Moreover, in the case of an unstable flow in 2D, significant saturation overshoot is observed in 1D. The most interesting is the intermediate case for fluxes between $q_{\text{top}} = 0.005 - 0.010 \,\text{cm}\,\text{min}^{-1}$. Surprisingly, a very small saturation overshoot is observed in 1D. Hence, even a small saturation overshoot indicates that the flow is preferential in 2D. This fine detail is very well captured by the semi-continuum model. Note that experimental measurements of such a small saturation overshoot are quite limited (e.g., using neutronography (Pražák et al., 1990)), since the magnitude of the saturation overshoot is between

$0.007 - 0.020$.





According to the stability analysis (Saffman and Taylor, 1958; Parlange and Hill, 1976), the flow should be stable if the influx is at least equal to the saturated conductivity $K_S$. This condition holds for the semi-continuum model regardless of the initial conditions. This is due to the fact that for two fully saturated blocks, the flux given by Eq. (6) is equal to $K_S = 15 \, \text{cm} \, \text{min}^{-1}$, because the pressure difference is zero in this case. Consequently, the flow will always be stable as long as the influx $q_{\text{top}} \geq K_S$

because all blocks of the porous medium will be fully saturated, hence saturation overshoot cannot occur. This is consistent with performed 1D and 2D simulations, where the flow is stable for $q_{\text{top}} = K_S$, as can be clearly seen in Fig. 10 and Fig. 5.

## 4 Discussion

The development of stable and unstable flows for the various applied fluxes is a complex problem that has been of interest for several decades (Saffman and Taylor, 1958; Parlange and Hill, 1976; Glass et al., 1989c; Yao and Hendrickx, 1996; DiCarlo,

2004). As the influx increases, the flow changes from stable to unstable, and back to stable again. It is a challenge to find a model that is able to simulate this complex transition along with finger width and finger spacing (DiCarlo, 2013). One of the most promising attempts was proposed by Beljadid et al. (2020), who developed a nonlocal model endowed with an entropy function. This model includes a fourth order spatial derivative of the saturation and is an extension of the Richards' equation. They showed the transition of an unstable to a stable wetting front for large applied fluxes. Moreover, a good agreement of the

finger width between simulations and experiments of Glass et al. (1989c) for large applied fluxes was demonstrated. However, for very low fluxes, the wetting front never becomes stable again, which is not consistent with the experiments (Yao and Hendrickx, 1996). On the contrary, decreasing infiltration rate leads to thinner fingers and a stable wetting front is hence not developed. Regardless of this discrepancy, we consider the results beneficial.

To the best of our knowledge, the semi-continuum model is the first model that is able to correctly predict stable and unstable

wetting fronts depending on the influx together with finger width and finger spacing (see Fig. 5 and Fig. 6). Even the range of the number of fingers in the simulations is consistent with the experiments of Glass et al. (1989c). Moreover, it is demonstrated in Fig. 9 that the flow is highly preferential in the case of unstable wetting front and becomes non-preferential in the case of stable wetting front. Finally, the hypothesis that the cause of the wetting front instability is the saturation overshoot (DiCarlo, 2013) is consistent with the simulations, given a match between 1D saturation overshoot and 2D wetting front instability. In

addition to the good agreement with laboratory experiments, two surprising features are found:

1. The flow can be preferential at low infiltration fluxes even when the porous medium is fully wetted and the water flows through the entire porous medium. This behavior is demonstrated in Fig. 8 for $q_{\text{top}} = 0.0005 \, \text{cm} \, \text{min}^{-1}$. The flow is still preferential because preferential pathways (a slight increase in saturation) arise during infiltration into a dry porous medium and these do not disappear over time, despite the porous medium is fully wetted at this time. And this slight

increase in saturation is enough to make the flow preferential, so water flows faster through these pathways. This is simply due to the relative permeability function, which is power-law in nature (see Eq. (5)). In left panel of Fig. 11, the by-pass ratio along with the saturation and the relative permeability at a depth of $30 \, \text{cm}$ is plotted. The highest values of the by-pass ratio correspond to the highest values of the relative permeability. Moreover, the effective permeability (the



multiplication of relative and intrinsic permeabilities) exactly follows the by-pass ratio, which is demonstrated in right

panel of Fig. 11.

2. Next feature is the connection between the saturation overshoot and the wetting front instability. It is shown in right
   panel of Fig. 10 that a very small saturation overshoot indicates the instability of the flow. This is exactly the case of
   $q_{\mathrm{top}} = 0.0005\,\mathrm{cm\,min}^{-1}$, where the magnitude of the saturation overshoot equal to $0.007$. It means that there is a rather
   close relationship between saturation overshoot in 1D and preferential flow in 2D. Moreover, the transition between

preferential and non-preferential flow is quite gradual as the influx decreases.

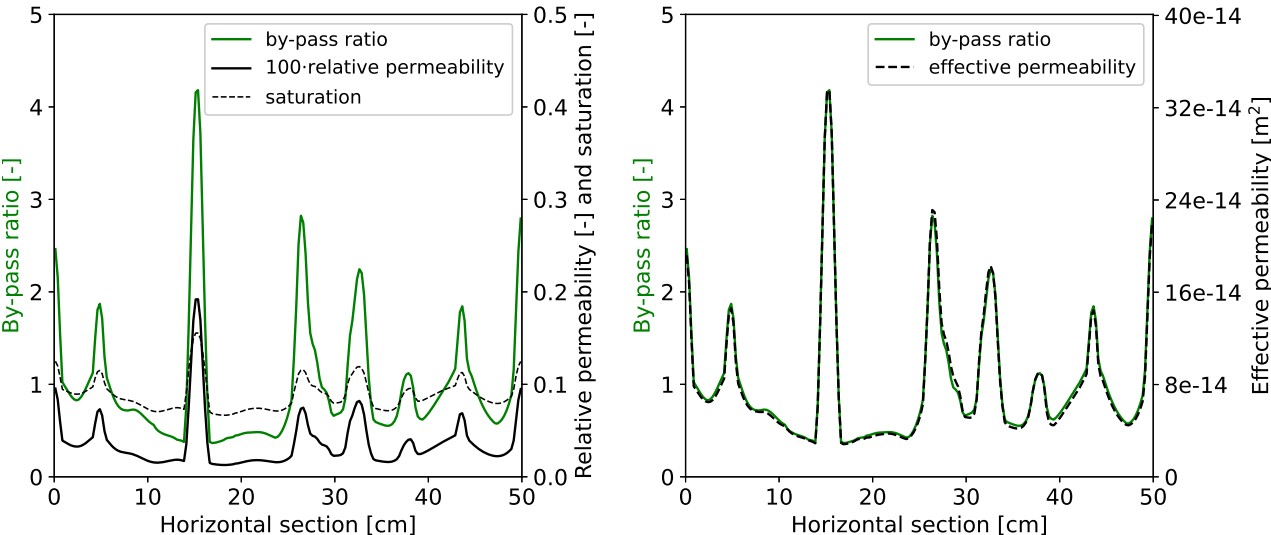

**Figure 11.** Analysis of preferential flow at low infiltration rates. Left panel: The by-pass ratio at a depth of $30\,\mathrm{cm}$ of the porous medium is plotted for $q_{\mathrm{top}} = 0.005\,\mathrm{cm\,min}^{-1}$ at time $t = 24000\,\mathrm{s}$ together with the saturation and the corresponding relative permeability. By-pass ratio values are plotted on the left $y$-axis and values of saturation and relative permeability are plotted on the right $y$-axis. Relative permeability values are multiplied by 100, since these values are much smaller compared to the saturation. It can be clearly seen that by-pass ratio values correlate with relative permeability values. Right panel: The by-pass ratio at a depth of $30\,\mathrm{cm}$ of the porous medium is plotted for $q_{\mathrm{top}} = 0.005\,\mathrm{cm\,min}^{-1}$ at time $t = 24000\,\mathrm{s}$ together with the effective permeability. By-pass ratio values are plotted on the left $y$-axis and values of the effective permeability are plotted on the right $y$-axis. Effective permeability values exactly follow the values of the by-pass ratio.

It has been shown mathematically and experimentally that the stability of the 2D wetting front correlates with the saturation overshoot in 1D (DiCarlo, 2013). Moreover, the semi-continuum model implies the same conclusions as was demonstrated in Sect. 3.6 *Wetting front instability*. To understand why the wetting front is stabilized at either low or high infiltration rates, it is therefore sufficient to study the origin of the saturation overshoot in 1D. Borrowing the term "hold-back pile-up effect" from

Eliassi and Glass (2001), the saturation overshoot occurs when the water cannot enter the dry porous medium and therefore water is held back. This hold back causes water to pile up above the interface between the wet and dry porous medium. As the





amount of water increases, the pressure gradient between the wet and dry parts of the porous medium increases as well and water can advance. We conjecture that this "hold-back pile-up effect" is mainly due to the very low conductivity between the wet and dry parts, because it is determined by the geometric or harmonic mean of the respective conductivities of both parts

(Jang et al., 2011). This is why the geometric mean is included in the numerical scheme of the semi-continuum model. Note that this explanation of pile-up mechanism also applies to the initial saturation dependence. It was experimentally found that as the initial saturation increases, the flow tends to be more diffusive (Bauters et al., 2000). Since at a sufficiently large initial saturation the resulting conductivity is high enough, the saturation overshoot does not occur.

Based on the above explanation, we can understand why the wetting front stabilizes at low infiltrations. When a small amount

of water flows into a dry porous medium, the conductivity of this part increases rapidly. This is due to the power law nature of the relative permeability function. Using the parameters in Table 1, when the saturation increases from $0.01$ to $0.05$, the relative permeability increases approximately $170$ times. At very low infiltrations, the water has enough time to proceed downwards, so that the conductivity of the dry part of the porous medium increases significantly, which subsequently allows more water to flow downwards. In principal, it means that the water does not have sufficient time to pile-up at low infiltration rates. On the

other hand, at very high infiltrations, the porous medium is fully saturated and hence the saturation overshoot cannot occur. This is well observed in Fig. 10 for $q_{\text{top}} = 15\,\text{cm}\,\text{min}^{-1}$, where the saturation is equal to one everywhere. As the applied flux decreases, the porous medium is not fully saturated and there is room for the saturation overshoot to arise. This can be seen for $q_{\text{top}} = 5\,\text{cm}\,\text{min}^{-1}$; the saturation at the finger tip is still equal to one, but the influx is not high enough to fully saturate the entire porous medium and the saturation overshoot develops. This observation is in an agreement with the experiments of

DiCarlo (2010). The author showed that the saturation of the finger tip approaches unity at very high infiltration fluxes for which a stable flow is observed (see figure 3 in DiCarlo (2010) for details).

Using the Darcy-Buckingham law given by Eq. (6) we can calculate the saturation in the finger tail. The saturation and pressure in the finger tail are constant because the flux between the blocks is stabilized and is given by the value of top boundary flux $q_{\text{top}}$. Hence, Eq. (6) implies:

$$k(S_{\text{tail}}) = \frac{q_{\text{top}}}{K_S}, \tag{8}$$

where $S_{\text{tail}}$ denotes the saturation in the finger tail within the relative permeability function. If the saturation in the block increases above $S_{\text{tail}}$, then the saturation overshoot will develop (water will pile-up). This is because the flow tends to stabilize, so the saturation of the block drops to $S_{\text{tail}}$ over time. In addition, the calculation of $S_{\text{tail}}$ is independent on the initial saturation, which is in agreement with the experimental measurements (Fritz, 2012) and also confirmed by the semi-continuum model

(Kmec et al., 2019). Moreover, for $q_{\text{top}} = K_S$, it follows from the Eq. (8) that $S_{\text{tail}}$ equals to one. This means that the saturation overshoot cannot occur because the porous medium is fully saturated in this case. This is consistent with the stability condition (Saffman and Taylor, 1958; Parlange and Hill, 1976; DiCarlo, 2013), which predicts the flow to be stable for $q_{\text{top}} \geq K_S$.

We remark that the values $S_{\text{tail}}$ for all top boundary fluxes used in the simulations correspond to the values of tail saturation plotted in right panel of Fig. 10 with an average difference of $5 \times 10^{-4}$. There are, however, two exceptions for

$q_{\text{top}} = 7.5\,\text{cm}\,\text{min}^{-1}$ and $q_{\text{top}} = 10\,\text{cm}\,\text{min}^{-1}$, for which the porous medium is fully saturated and therefore the saturation



overshoot cannot develop. Note that if the saturation overshoot occurs, e.g., when using a higher initial saturation, then the saturation in the finger tail will be given by Eq. (8). This behavior is confirmed in Appendix A3.

The measurement of macroscopic properties of the homogeneous porous medium, such as permeability or porosity, is usually performed by averaging the microscopic quantities over the domain (White et al., 2006; Ghanbarian et al., 2021). For more

realistic simulations, it is appropriate to use the spatial variation of the continuum quantity. Therefore, a small distribution of the intrinsic permeability was used in simulations. One may wonder whether the obtained results depend on the used distribution. In Kmec et al. (2023) it was demonstrated that it has no significant effect on the flow, because the nature of the flow remains the same even when eight different intrinsic permeability distributions were used.

The semi-continuum model was shown to be consistent with the experiments in 1D (Kmec et al., 2019) and 2D (Kmec et al.,

2021, 2023). Moreover, taking into account the results presented in this paper, we conjecture that the model has been validated by the core experiments performed in an unsaturated homogeneous porous medium (Glass et al., 1988, 1989b, c, a; Selker et al., 1992; Liu et al., 1994; Yao and Hendrickx, 1996; Bauters et al., 2000; DiCarlo, 2004; Sililo and Tellam, 2005; Rezanezhad et al., 2006; DiCarlo, 2007, 2010). In the case of heterogeneous porous medium, the model was already used (Kmec et al., 2021) to simulate water infiltration experiments into a layered porous medium (Rezanezhad et al., 2006). There are other well-

conducted experimental works of flow into heterogeneous porous media, e.g., by Cremer et al. (2017). Therefore, using the model to simulate flow in a heterogeneous porous medium may provide another opportunity for its validation.

## 5  Conclusions

It has long been known that the wetting front in an initially dry and homogeneous porous medium depends strongly on the applied infiltration flux. The wetting front is observed to be stable for low and high infiltration fluxes and is unstable within a

certain range of flux. The instability of the wetting front is manifested by the formation of fingers that vary in width, velocity and spacing. Although experiments have been known for decades, no model has yet been developed to reliably capture this dependence. In this paper, the governing equation containing a Prandtl-type hysteresis operator under the spatial derivative is introduced. It is a formal limit of the semi-continuum model, which is the model used to simulate flow in porous media. It is shown that the semi-continuum model correctly captures the complex behavior of infiltration flux dependence. This includes

the transition from stable to unstable and back to stable flow as the infiltration flux increases along with calculating the correct finger width and finger spacing. In addition, the model helps explain preferential flow and understand the formation of an unstable wetting front in terms of the saturation overshoot.



## Appendix A

### A1  Dependence of flow on infiltration rate

All $18$ simulations for infiltration rates in the range $q_{\text{top}} = 0.001 - 15\,\text{cm}\,\text{min}^{-1}$ are presented here. Figures A1 and A2 show a snapshot of the saturation profiles for all the applied fluxes. The time for each influx is chosen so that the saturation reaches $40\,\text{cm}$ from the upper boundary. The applied influx and simulation time are displayed for each frame at the upper left corner. The transition between stable and unstable flow is well observed.

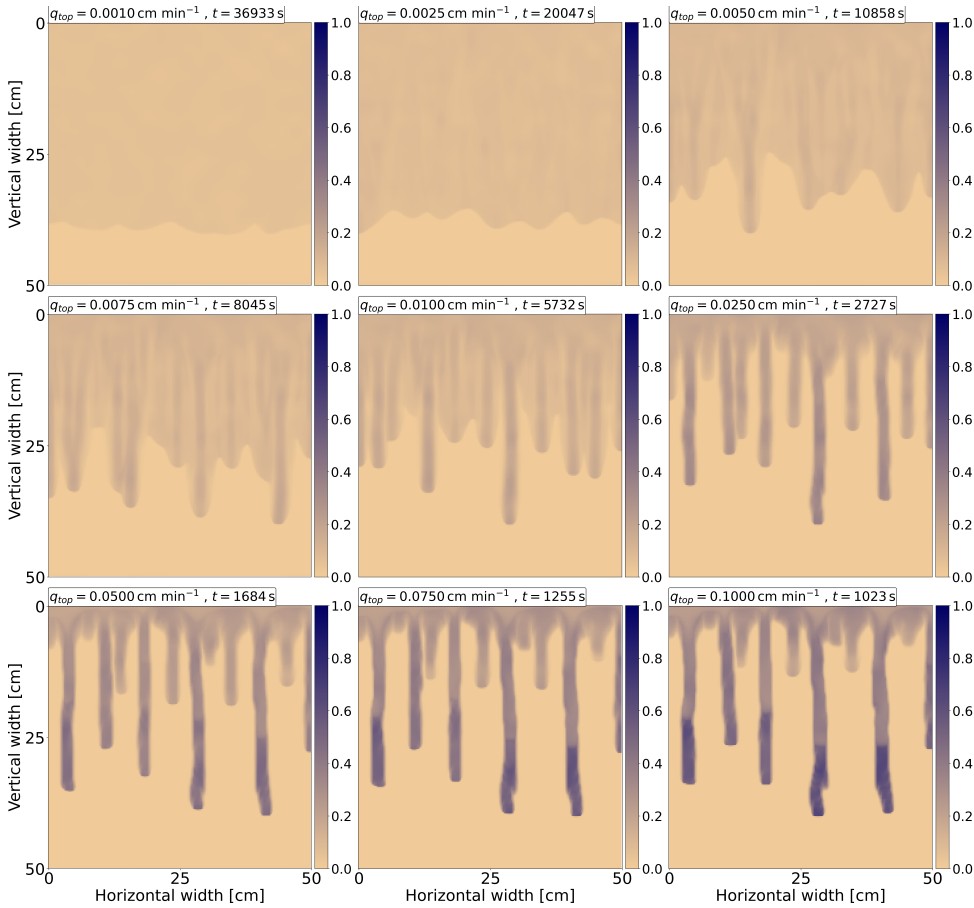

**Figure A1.** Saturation profiles for nine different infiltration rates from $q_{\text{top}} = 0.001\,\text{cm}\,\text{min}^{-1}$ to $0.1\,\text{cm}\,\text{min}^{-1}$. For each frame, the influx is displayed together with the simulation time. Saturation is colored according to the color bar on the right.



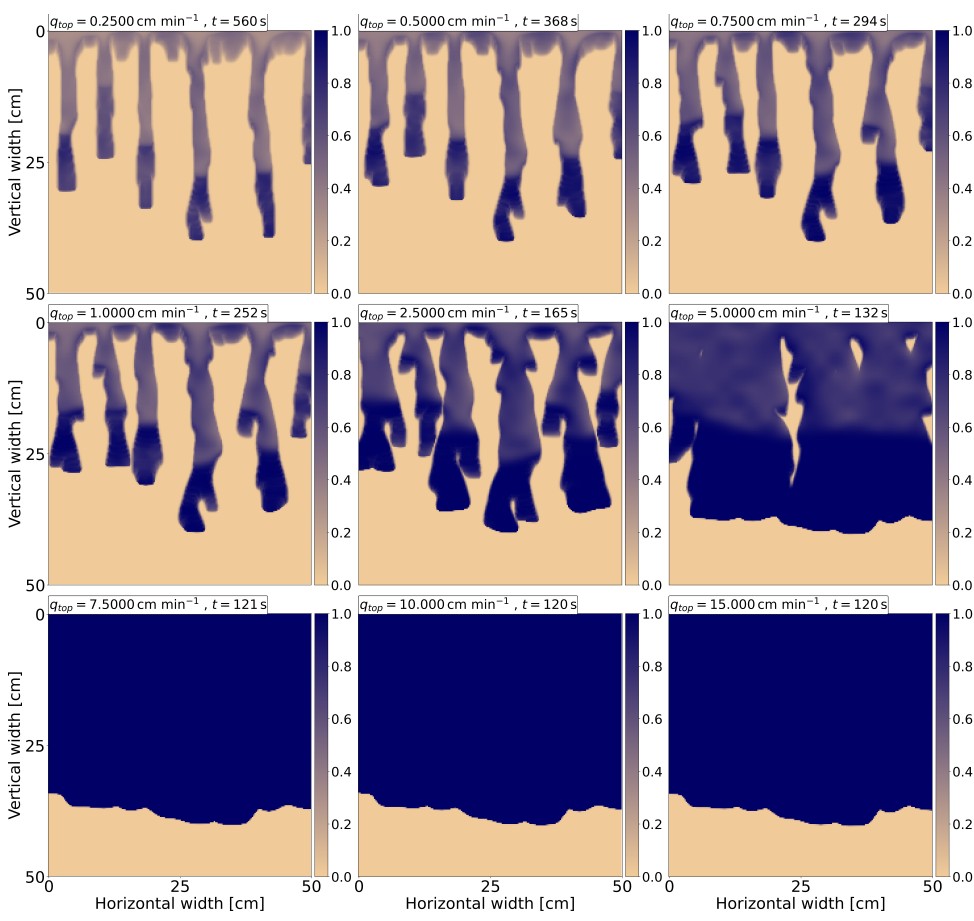

**Figure A2.** Saturation profiles for nine different infiltration rates from $q_{\text{top}} = 0.25\,\text{cm}\,\text{min}^{-1}$ to $15\,\text{cm}\,\text{min}^{-1}$. For each frame, the influx is displayed together with the simulation time. Saturation is colored according to the color bar on the right.



## A2 Finger segmentations

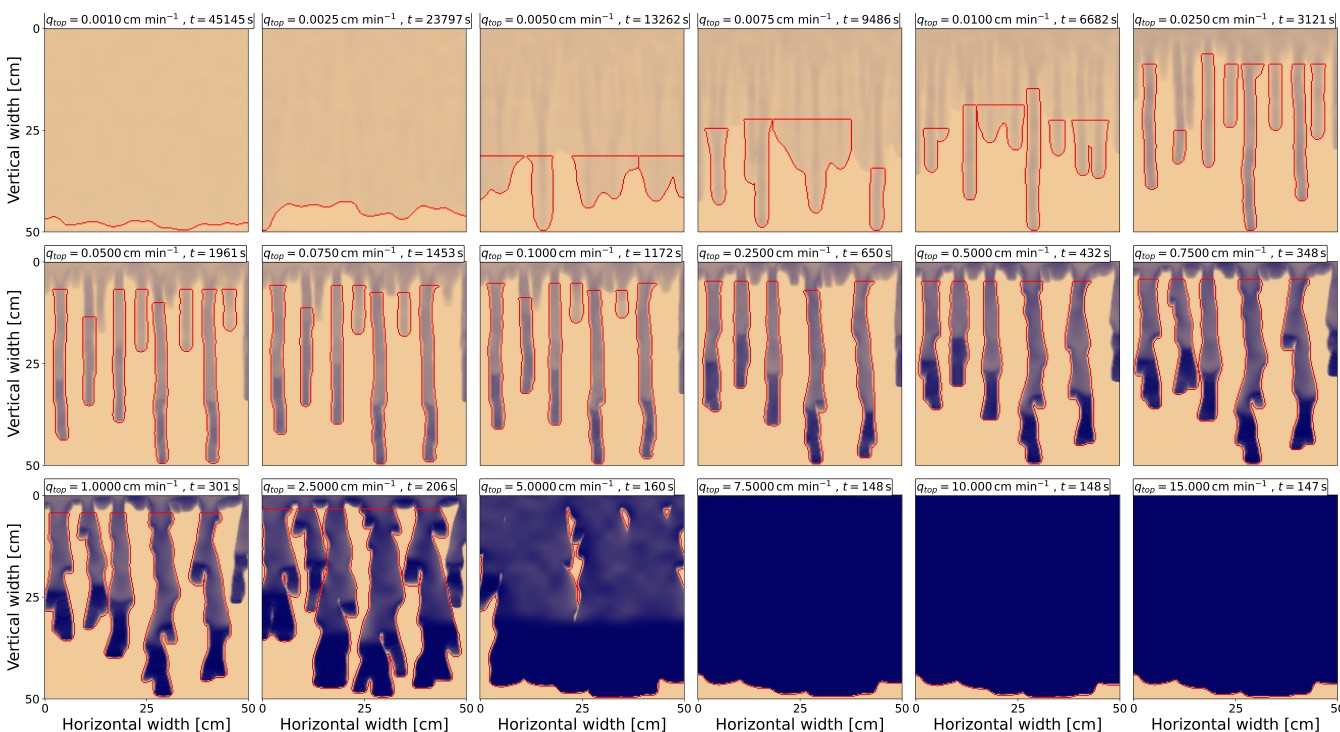

**Figure A3.** Finger segmentations for 18 different infiltration rates in the range $q_{\text{top}} = 0.001 - 15 \, \text{cm} \, \text{min}^{-1}$. The segmentations are marked with a red color. The time for each influx is chosen so that the saturation reaches the bottom of the chamber, i.e., $50 \, \text{cm}$. The applied influx and simulation time are displayed for each frame at the upper left corner.

## A3 By-pass ratio

Figure A4 shows the by-pass ratio at a depth of $30 \, \text{cm}$ of the porous medium for 18 different infiltration rates in the range $q_{\text{top}} = 0.001 - 15 \, \text{cm} \, \text{min}^{-1}$. The by-pass ratio is marked in red and is plotted together with the saturation across the same horizontal section. Saturation values are marked in blue. It can be clearly seen that the highest values of the by-pass ratio correlate with the location of the highest saturation values. However, the correlation between the saturation and the by-pass

ratio is lower if the saturation is high enough. This is evident for $q_{\text{top}} \geq 7.5 \, \text{cm} \, \text{min}^{-1}$. In this case, a slight change in saturation no longer significantly changes the relative permeability values and so does not change the by-pass ratio. Instead, the intrinsic permeability becomes the dominant factor of the flow.

Note that the porous medium is first fully saturated for $q_{\text{top}} = 7.5 \, \text{cm} \, \text{min}^{-1}$ and $q_{\text{top}} = 10 \, \text{cm} \, \text{min}^{-1}$ (see Fig. A2). However, the saturation then decreases because the bottom boundary flux $q_{\text{bot}}$ approximately equals to $K_S = 15 \, \text{cm} \, \text{min}^{-1}$ and is

thus larger than $q_{\text{top}}$. In this case, the saturation should correspond to the calculated value $S_{\text{tail}}$ given by Eq. (8). Considering



the distribution of the intrinsic permeability, the average value of the calculated $S_{\text{tail}}$ over all blocks at a depth of $30\,\text{cm}$ is $0.8540$ and $0.9218$ for $q_{\text{top}} = 7.5\,\text{cm}\,\text{min}^{-1}$ and $q_{\text{top}} = 10\,\text{cm}\,\text{min}^{-1}$, respectively. Since the average values from simulations at a depth of $30\,\text{cm}$ are $0.8552$ and $0.9225$, the saturation indeed corresponds to the calculated values $S_{\text{tail}}$. Therefore, the saturation at the finger tail always tends to decrease to the value obtained by Eq. (8). Moreover, for $q_{\text{top}} = 15\,\text{cm}\,\text{min}^{-1}$, the

porous medium is still fully saturated even if water flows from the bottom boundary. This is due to the fact that $S_{\text{tail}} = 1$ in this case.

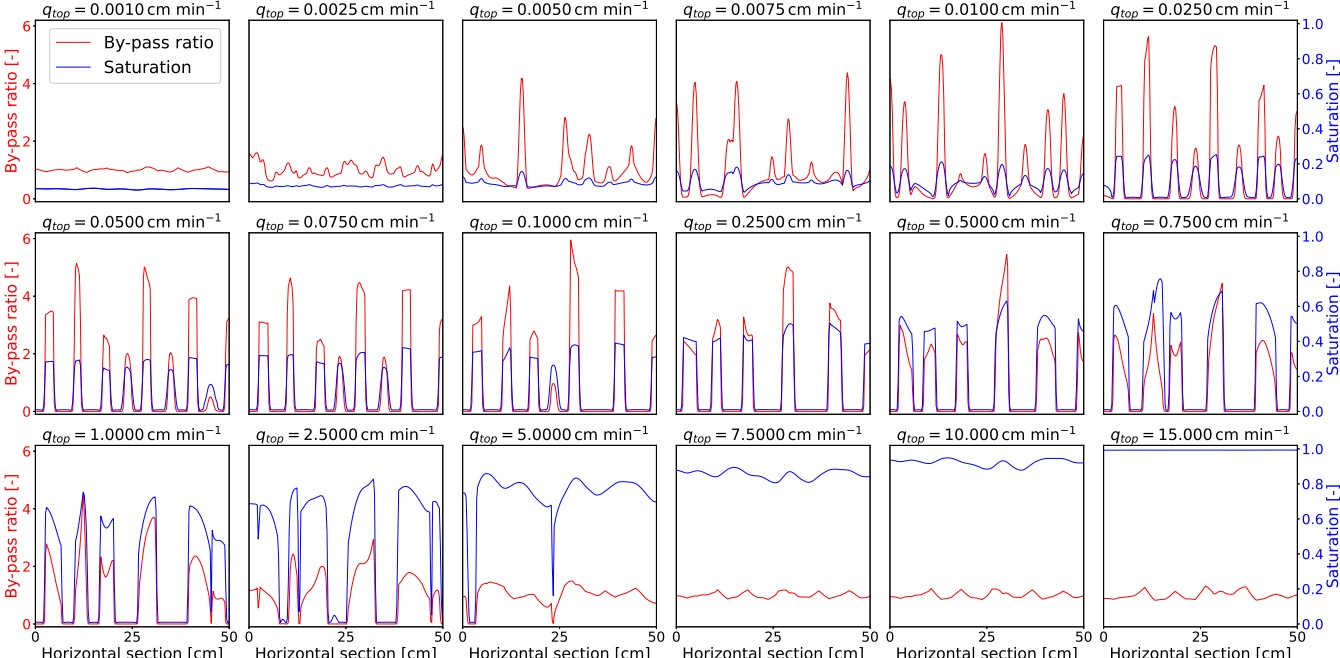

**Figure A4.** The by-pass ratio at a depth of $30\,\text{cm}$ of the porous medium for 18 different infiltration rates in the range $q_{\text{top}} = 0.001 - 15\,\text{cm}\,\text{min}^{-1}$. The values of by-pass ratio are marked in red. Together with the by-pass ratio, the saturation (blue lines) is plotted across the same horizontal section, i.e., at a depth of $30\,\text{cm}$ of the porous medium. By-pass ratio values are plotted on the left $y$-axis and values of saturation are plotted on the right $y$-axis. For each frame, the corresponding influx is shown at the top.



*Code and data availability.* The software code that produced the simulations is written in Python and can be downloaded from Kmec (2023a). The code can be used for 1D, 2D and 3D simulations. Simulation data that are needed to create the plots included in the manuscript can be downloaded from Kmec and Šír (2023a, b). Please do not hesitate to contact us if you encounter any problems when downloading the software code and simulation data.

*Video supplement.* Videos of the transient simulations of all 2D simulations can be downloaded from Kmec (2023b).

*Author contributions.* JK and MS wrote the manuscript, JK implemented the computer code, ran and analysed the simulations.

*Competing interests.* The authors declare that they have no conflict of interest.

*Acknowledgements.* Jakub Kmec gratefully acknowledges the support by the Operational Programme Research, Development and Education, project no. CZ.02.1.01/0.0/0.0/17_049/0008422 of the Ministry of Education, Youth and Sports of the Czech Republic. Computational resources were provided by the e-INFRA CZ project (ID:90254), supported by the Ministry of Education, Youth and Sports of the Czech Republic. Special thanks goes to Rostislav Vodák for helpful discussion of the theoretical part of this paper and to Tadeáš Fryčák for his significant help in improving the code of the semi-continuum model.



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
