# Peer review of "Modeling 2D gravity-driven flow in unsaturated porous media for different infiltration rates"

_EGUsphere, 2023_

## Referee Comment (RC2)

[referee-annotated manuscript omitted]

---

## Author Comment (AC1)

Dear reviewer, we appreciate your comments. We provide a detailed answers below in the form of bullet points.

Reviewer comment: The manuscript deals with a classical 1D analysis of the unsaturated flow stability during a wetting event.

- It is important to note that the manuscript is not focused on 1D analysis, but 2D analysis instead. Specifically, we show how 2D flow varies with different inflows changing from diffusion flow to fingering flow and back to diffusion flow as the infiltration rate increases (viz results sections 3.1 – 3.5 and discussion section 4).

  In the case of 1D flow, there exist a few models that are able to capture the transition between diffusion and finger-like behavior for various infiltration rates, for instance [Cueto-Felgueroso, Juanes, 2009]. However, the 2D case is significantly different because not only this transition but also finger spacing and finger width are addressed [DiCarlo, 2009]. This makes the analysis much more complex and complicated. One of the main evaluating factors according to DiCarlo is that the model "can produce predictions of the 2-D and 3-D preferential flow in terms of finger widths and finger spacings." In our manuscript, we have demonstrated that the semi-continuum model is capable of this. Moreover, to the best of our knowledge, no such model has yet been available to simulate this complex 2D behavior.

  Let us note that all simulations performed in the manuscript are in 2D. The exception is section *3.6 Wetting front instability*, where we analyze the connection between 1D and 2D flow in unsaturated porous media. This section is included to demonstrate that we are consistent with experimental observations, so that the saturation overshoot in 1D is closely related to the preferential flow in 2D. Again, to the best of our knowledge, such a connection has not been demonstrated by any model so far.

  To make this as clear as possible, we will highlight it in the manuscript in more detail. And we also plan to change the title of the manuscript to make it clear that we are dealing with 2D simulations.

Reviewer comment: The numerical treatment of the problem at stake, although quite standard and therefore with no new insight, appears to be correct (I went through it, and I didn't find any error).

- The numerical discretization of the governing equation is indeed standard with one major exception. The discretization of the Prandtl-type hysteresis operator is not common in flow modeling as the discretization is dependent on the size of the used mesh (blocks). However, this is not a key part of the manuscript as the semi-continuum model has been already published, e.g., in [Vodák et al., 2022] and [Kmec et al., 2023].

Reviewer comment: Instead, the English usage requires a very solid and sounding proofreading.

- English will be proofread for the revised manuscript.

Reviewer comment: Besides this (marginal) aspect, the main issue which I see with the manus is of methodological nature. In particular, my skepticism is two-fold. First, accounting for the gravity solely (thus neglecting the impact of retention) may be unrealistic especially if one is interested (as it is usually happens in the applications) on the "onset" of the stability vsinstability".

- The semi-continuum model describes the flow of water in unsaturated porous medium. Water is characterized by saturation in the equation, since saturation is defined as the ratio of the total water volume to the pore volume. This is a standard approach in modeling multiphase flow [DiCarlo, 2009].

  Note that the gravity is essential in forming the finger flow and is therefore included in the semi-continuum model. However, retention is not neglected at all; the water retention is controlled mostly by the pressure-saturation relation (the water retention curve) defined by equations Eq. (2) and Eq. (3). Moreover, according to Eq. (6), the gravity term takes a role only in the case of vertical fluxes, while the term related to the retention curve is included for horizontal fluxes as well. Again, this approach is standard to model water retention in porous medium, for example using the Richards' equation [Simunek, 1994].

Reviewer comment: *Second, the authors have carried out a long and intensive analysis of the flow rates which make (or not) stable the flow, but what and where is the stability analysis?*

- There is probably a misunderstanding in the used terminology and we apologize for that. The stability refers to diffusion-like flow where no saturation overshoot is observed, and the instability refers to preferential flow (or finger-like flow) that is accompanied by the saturation overshoot. This is standard terminology used in the community [DiCarlo, 2013]. However, to make it as clear as possible, we will explain it the manuscript and change the text accordingly. Also, we plan to change the title of the manuscript to: *Modeling 2D gravity-driven flow in unsaturated porous media for different infiltration rates.*

  A short part of the manuscript deals with the instability of the wetting front (section *3.6 Wetting front instability*). As mentioned in the response to the first reviewer's comment, in this section we analyze the connection between saturation overshoot in 1D and preferential flow in 2D. However, this is not the main part of the manuscript because we focus on 2D preferential flow and show that the semi-continuum model correctly captures flow experiments that no model has captured before.

Reviewer comment: *The very new and innovative insight could have been a Touring analysis of the stable/unstable flow patterns in order to highlight which ones are those parameters (and perhaps the infiltration rate is the most important one) that regulate such a stability. Instead, the manuscript, as it is, is nothing more a numerical analyses (followed by an experimental benchmark), quite similar to many others, already existing in the literature.*

- Thank you for your suggestion to include analysis using Turing instability and Turing pattern concept. However, the manuscript addresses different topic as the stability/instability has different meaning (see the response to the previous comment). It deals with 2D preferential flow as a function of applied influx. Specifically, we show that the model is able to predict 2D preferential flow in terms of finger widths and finger spacings.

  According to "quite similar to many others, already existing in the literature": We conjecture that there are no other models that are able to correctly capture the transition from diffusion to finger-like and back to diffusion flow as the infiltration flux increases. In our humble opinion, the most promising model is a nonlocal model proposed by [Beljadid et al., 2020], however, the flow is preferential for very low fluxes which is not consistent with experiments. This is not the case for the semi-continuum model presented in our manuscript. The model correctly captures the influx dependence even for very low fluxes and, in addition, the finger widths and finger spacings are consistent with experimental observations, as shown in section *3.4 Finger width as function of influx.*

  However, if there is another model dealing with the same topic (i.e., 2D preferential flow as a function of applied influx), we would like to ask the reviewer for a reference, where this issue is properly addressed. We would very much appreciate that.

**References**

[DiCarlo, 2009], D. A. DiCarlo, Can continuum extensions to multiphase flow models describe preferential flow? Vadose Zone J., DOI: 10.2136/vzj2009.0099

[Simunek, 1994], J. Šimůnek and D. L. Suarez, Two-dimensional transport model for variably saturated porous media with major ion chemistry, Water Resour. Res., DOI: 10.1029/93WR03347

[DiCarlo, 2013], D. A. DiCarlo, Stability of gravity-driven multiphase flow in porous media: 40 Years of advancements, Water Resour. Res., DOI: 10.1002/wrcr.20359

[Cueto-Felgueroso, Juanes, 2009], L. Cueto-Felgueroso and R. Juanes, A phase field model of unsaturated flow, Water Resour. Res., DOI: 10.1029/2009WR007945

[Vodák et al., 2022], R. Vodák and T. Furst and M. Šír and J. Kmec, The difference between semi-continuum model and Richards' equation for unsaturated porous media flow, Scient. Rep., DOI: 10.1038/s41598-022-11437-9

[Kmec et al., 2023], J. Kmec and M. Šír and T. Furst and R. Vodák, Semi-continuum modeling of unsaturated porous media flow to explain Bauters' paradox, Hydrol. E. Sys. Sci., DOI: 10.5194/hess-27-1279-2023

[Beljadid et al., 2020], A. Beljadid and L. Cueto-Felgueroso and R. Juanes, A continuum model of unstable infiltration in porous media endowed with an entropy function, Adv. Water Resour., DOI: 10.1016/j.advwatres.2020.103684

---

## Author Response (AR1)

Dear editor and reviewers, please find below a point-by-point response to all the comments raised during the review process. The response structure is as follows: (1) comments from reviewers, (2) the author's previous responses to reviewer's comments, and (3) the author's changes in the manuscript. Before proceeding to the reviewer's comments and corresponding changes in the manuscript, we first provide general comments below.

- We first addressed comments from reviewers #2 and #3, as we find their comments the most relevant. Following that, we addressed comments from reviewer #1.

- Comments from reviewers are highlighted in blue, and the previous author's responses are in black to maintain consistency with the previous review process. For clarity, comments indicating changes in the manuscript are highlighted in red.

- All important changes are marked in blue in the marked-up revised manuscript, except for changes related to the English language, which has been revised by authors and subsequently proofread by an English speaker.

- We have divided the Discussion section into separate subsections to improve its clarity. Namely:

    - 4.1 Preferential flow in the case of fully wetted porous medium
    - 4.2 Formation of saturation overshoot and stabilization of wetting front
    - 4.3 Importance of the geometric mean in terms of water entry value
    - 4.4 Effect of intrinsic permeability distribution
    - 4.5 Model validation and future plans

- The discussion during the review process was extensive, resulting in inclusion of many relevant points in the revised manuscript. This includes, for instance, improved motivation for the manuscript, a detailed section on the Prandtl-type hysteresis operator (Sect. 2.1), and sections 4.3 and 4.4 discussing the importance of the geometric mean and the effect of intrinsic permeability distribution.

    To avoid an unnecessary increase in the manuscript's length due to the inclusion of new text, we have shortened some extensive parts of the manuscript.

- We have included all our suggestions raised during the review process into the manuscript.

**Reviewer #2**

**Part 1**

Reviewer's comments: `https://doi.org/10.5194/egusphere-2023-2785-RC2`
Author's response: `https://doi.org/10.5194/egusphere-2023-2785-AC2`

Before proceeding to the reviewer's comments, we would like to acknowledge that this research on soil water flow has been inspired by the work of Kutílek and Nielsen [1]. They argue that a correct hydrologic-scale rainfall-runoff model should incorporate a realistic small-scale model of infiltration, retention, and runoff into the geological bedrock. Many Czech and Slovak hydrologists (including e.g., J. Šimůnek) were students of M. Kutílek and his colleague M. Císlerová, who all have been deeply involved in the study of the coexistence of preferential and diffusive flow in soil [2, 3]. At present, the third generation of researchers following ideas of M. Kutílek is actively involved in fulfilling this research program, focusing on the development of a fully realistic small-scale model, which we believe is beneficial for hydrological modeling. Since the motivation of our manuscript is not well explained, we intend to improve it in the Introduction section. Please see also our responses to the reviewer's comments n. 1, n. 3 and n. 9 for further details.

The motivation of the manuscript has been improved. Since the changes related to the manuscript motivation are discussed extensively in several points below, we chose to summarize all the changes here. This includes all changes from the Introduction section at lines 13–110.

- We have briefly introduced the gravity-driven multiphase flow, along with new additional references.

- We have clarified and slightly extended the previous text on diffusion-like and finger-like flow with emphasis on the recommendation that rainfall-runoff models should include a robust model of soil water movement.

- The literature related content has been moved from the Results and Discussion sections into the Introduction. Consequently, the previous corresponding text from the Introduction section has been slightly extended and clarified.

- We have provided a brief overview of microscale and macroscale models, highlighting respective advantages and disadvantages.

- We have included why small-scale validation of microscale models is necessary before moving to field-scale simulations. Moreover, we have addressed a small-scale validation procedure based on DiCarlo's approach.

- We have discussed the importance of experiments of Glass et al. [4] and Yao and Hendrickx [5].

Reviewer comment n. 1: The paper introduces a hysteretic model that can generate unstable infiltration fronts and the resulting fingering patterns. The model is tested by reproducing highly idealized experiments in two-dimensional sand tanks, with very limited additional analysis.

- Although some of the experiments have been performed in Hele-Shaw cells ("two-dimensional tanks"), they have not yet been successfully reproduced by any published model, even though these experiments are over 30 years old. We believe that demonstrating the ability of the semi-continuum model to reproduce experiments with relatively complex behavior is highly beneficial.

  We propose that we better explain our motivation in the manuscript, and why we chose to simulate 2D infiltration experiments. Our goal is to fully validate the semi-continuum model with well-known laboratory experiments, following DiCarlo's approach [6]. DiCarlo suggested evaluating a model, which consists of four points (see section 6.6 in [6]). The only point the semi-continuum model has not fully addressed is the fourth one, which states that the model "Can produce predictions of the 2-D and 3-D preferential flow in terms of finger widths and finger spacings." We have already achieved successful reproduction of the dependency on initial saturation [7, 8]. However, the final aspect required for full validation is to accurately capture the dependence on the infiltration rate of 2D/3D experiments [5, 4, 6]. Furthermore, ongoing efforts within the community, as evidenced by Beljadid et al. [9], emphasize the continued interest in reproducing these experiments. We plan to improve our motivation in the manuscript. Details are also given in other responses below.

  This point is related to the manuscript motivation and is thus closely tied to all changes performed in the Introduction section. Specific explanation of why we chose these experiments can be found on lines 79–105.

Reviewer comment n. 2: The model and its equations are not terribly well explained, which creates some confusion. The model test on experimental data is convincing, but there is no follow up. There is no indication if the model is of any use to apply to natural conditions or to address real world problems. This gives the impression of only half a paper: the presentation of a model and its validation, but no meaningful application.

- The reader is typically not familiar with a partial differential inequality, which is characterized in the model by a Prandtl-type hysteresis operator. However, we agree that we need to improve the explanation, and we plan to address this according to the reviewer's comments in the annotated PDF file.

  The crucial misunderstanding is regarding to the Prandtl-type hysteresis operator [10] utilized in the semi-continuum model. Contrary to the reviewer's suggestion, we are not presenting a nonideal relay contained in the Preisach model of hysteresis. To improve clarity, we have chosen to modify Figure 1 in the manuscript to better illustrate the Prandtl-type hysteresis model. Moreover, given that the Prandtl-type hysteresis operator provided by Eq. (1a) is not widely used in the soil science community, we plan to explain it before introducing the governing equation. Please find the modified Figure 1 of the Prandtl-type hysteresis operator below. In this figure, blue lines represent the limits of main wetting and draining branches, while black lines represent non-vertical scanning curves. The mathematical description of the Prandtl-type hysteresis operator can be found in Visintin [10] on page 16 (eqs. 2.3 - 2.5); see also the corresponding Figure 3 on page 15. Further details regarding this point are given in our annotated PDF file.

[Figure]

Lines 117–135: The illustration of the Prandtl-type hysteresis operator has been modified (Figure 1 in the manuscript). Moreover, the Prandtl operator has been explained before introducing the governing equation in Sect. 2.1 *Prandtl-type hysteresis operator.*

- The model can also be used for more complex natural conditions by incorporating different initial and boundary conditions. To make it as clear as possible, we plan to discuss its applicability to large-scale in-field experiments in the Discussion section.

  Please note that we chose to first demonstrate the ability of the model to correctly replicate laboratory experiments. These experiments, despite their seemingly simple nature, produce non-intuitive and complex outputs. Further details are provided in the response to the reviewer's comment n. 3, which addresses the same concern.

  Lines 534–540: We have incorporated the model's applicability to large-scale experiments and discussed our future plans in Sect. 4.5 *Model validation and future plans.*

Reviewer comment n. 3: The model is not new, and the simulations cover a very limited scope, merely attempting to reproduce laboratory experiments under conditions that have no relevance for the field. This strongly constricts the overall contribution of the paper.

- Laboratory experiments provide a detailed analysis of the flow behavior in the porous medium and so the validation of the model becomes more challenging. The primary motivation is to demonstrate the capability of the semi-continuum model to accurately capture the transition between diffusion and preferential flow under various applied fluxes in 2D. To the best of our knowledge, this is the first case of a model that has achieved this, which we find particularly interesting and valuable.

We conjecture that the mechanism of the flow should be perfected on a small-scale (e.g., laboratory tanks) so that the results obtained are consistent with experiments. Without consistency with experiments at a small-scale, the use of such a model for a large-scale (large field experiments) becomes irrelevant. Therefore, we believe that, as a fundamental step, any model should initially replicate well-defined laboratory experiments with their complex behavior. Following rigorous validation, one can then transition to more complex in-field experiments. Hence, we disagree with the notion that the conditions used in laboratory experiments lack relevance for in-field experiments. On the contrary, we believe that well-defined boundary and initial conditions in laboratory experiments enable a thorough examination of simulation details, allowing us to determine whether the model accurately captures preferential and diffusion flow.

While the experimental setup may be simple, the outcomes remain counter-intuitive, and no other model has demonstrated the ability to simulate these experiments [6]. If a model cannot replicate even "simple" laboratory experiments, it raises questions about its suitability for handling more complex in-field scenarios. This is why we have chosen to comprehensively validate our model under different initial and boundary conditions, following the recommendations of DiCarlo [6]. Furthermore, when using in-field experiments for model validation, the lack of captured details makes it challenging to distinguish whether any perceived model agreement is a result of, for instance, over-fitting the model parameters; the following bullet point addresses this issue.

- Our conviction for thorough model validation on small-scale experiments is based on our past experience when we chose the opposite approach, namely utilizing a model for in-field experiments without proper small-scale validation. In [11], we utilized a model based on upscaling the Richards' equation with a modified retention curve that includes an air entry value. This model was used to simulate outflow from the Liz catchment covering an area of $1\,\text{km}^2$ during the vegetation season of 1999. The good agreement between measured and simulated runoff suggested that we had found a suitable physical model for soil water flow.

  However, subsequent 2D simulations using the same model showed that water always flows through the entire porous medium, even when saturation overshoot occurs. This is experimentally contradicted, as water typically flows through wetted pathways while much of the porous medium remains dry. In fact, we have performed 3D experiments demonstrating that even in a relatively homogeneous porous material, water flows through a completely heterogeneous set of wetted fingers [12]. In summary, despite successfully capturing the runoff from the catchment (a "real world problem"), this model still turned out to be incorrect. This previous experience has shown us that the model must be physically correct in the details (small-scale); otherwise, the upscaling would be fundamentally wrong. Therefore, we focused on developing a new approach to modeling the description of water flow in porous media, followed by a thorough experimental validation on a small-scale.

  Please note that we also plan to explain our motivation for prioritizing small-scale validation before moving to field experiments in the manuscript. This essentially follows our response to the reviewer's comment n. 1, specifically DiCarlo's approach to model evaluation.

  This reviewer comment and our associated responses are related to the manuscript motivation and are thus closely tied to all changes performed in the Introduction section.

Reviewer comment n. 4: Although the authors discuss the literature at length throughout the paper, they seem to have missed many relevant papers. In the detailed comments I provide numerous references that may be useful to turn this paper into material that deserves publication.

- Thank you for your suggestions that are relevant for our manuscript. We have thoroughly reviewed the provided references and plan to incorporate them into the manuscript.

  Lines 13–17, 200: We have incorporated all the provided references into the manuscript, but not Thwaites et al. to avoid extending the manuscript excessively.

Reviewer comment n. 5: The English needs to be improved a lot. I had a hard time understanding important sections of the paper. In combination with the inadequate explanation of the equations and their variables, it is not really possible to grasp the theoretical elements of the paper.

- We will improve the English, with emphasis on the highlighted parts in the annotated PDF that the reviewer found challenging to understand.

  The English has been revised by the authors and subsequently proofread by an English speaker.

Reviewer comment n. 6: The model as such is interesting, and could be part of an interesting paper if it is better explained. To make the paper complete, the model would have to be used to carry out simulations that have some relevance for the real world, not the world of 2D tanks filled with air-dry artificial porous media.

- We plan to improve the explanation of the semi-continuum model based on the comments provided in the reviewer's annotated PDF file.

  We have improved the explanation of the semi-continuum model in Sect. 2. Most of the changes are related to the reviewer's annotated PDF file. We do not repeat reviewer's comments from the annotated PDF here.

  - Some paragraphs have been moved to make the text more readable. For instance, various approaches to modeling hysteresis are now discussed in Sect. 2.4 *Discretization of the Prandtl-type hysteresis operator* instead of Sect. 2.2 *Governing equation.*

  - Lines 117–135: The illustration of the Prandtl-type hysteresis operator has been modified (Figure 1 in the manuscript). Moreover, the Prandtl operator has been explained before introducing the governing equation in Sect. 2.1 *Prandtl-type hysteresis operator.*

  - Lines 142, 144–148 and Eq. (2): We have clarified in the manuscript that $k(S^-)$ and $k(S^+)$ denote the left and right limits. Moreover, we have improved the notation of the governing equation and clarified the text accordingly.

  - Lines 149–152: We have clarified why the formal limit of the semi-continuum model is a hyperbolic-parabolic differential equation in the case of an unsaturated porous medium. The explanation how it affects the solution of the equation is on lines 245–250.

  - Lines 164–177: The beginning of Sect. 2.4 has been clarified.

  - Lines 185–187: We have explained that an inflection point can also be chosen in Eq. (4).

  - We have clarified the caption of Fig. 2.

  - Lines 191–197: We have clarified why main branches become flatter as the pore size variability decreases, and consequently take a form of step-like functions. Moreover, we have explained why it aligns with the proposed scaling of the retention curve. The text on lines 184–186 has been slightly modified.

  - Lines 198–202: The paragraph regarding various approaches to modeling hysteresis has been simplified.

  - Lines 229–230: We have rephrased the text regarding the notation of effective permeability.

  - Lines 234–235: We have explained that the geometric mean utilized in Eq. (2) aligns with Eq. (7).

  - Line 239: We have clarified that the reason why the flow may exceed one is not due to the non-implementation of an adaptive time step.

  - We have removed the paragraph on lines 180–183 in the first version to avoid excessive extension of the manuscript.

  - Lines 244–249: We have slightly modified the paragraph related to the difference between the Richards' equation and our model.

- We have previously discussed the relevance of the laboratory experiments to the real world in our earlier responses. However, it is important to stress out that the dependence of flow on precipitation intensity is crucial at the hydrological scale, where the precipitation-runoff relationship is the field of interest. Previously, it was assumed that rainfall of low intensity and small total amount would soak into the soil so that the water would be captured in a surface layer and would not infiltrate into the geological bedrock. Hence, it would not lead to significant outflow from the catchment. However, it has been demonstrated that this is not true, and even low-intensity rainfall can result in preferential flow that leads to water flowing to greater depths. Consequently, this can cause a significant outflow from the catchment. Therefore, from a hydrological point of view, it is crucial to understand how the water infiltrates into the porous medium under various precipitation intensities.

  Consequently, validation of the model for different influx rates is necessary [6]. This is why the community typically begins with validating models using 1D laboratory experiments to explore the dependence of infiltration rate [13]. Let us cite some examples with 1D validations [14, 15, 16, 17]. Naturally, the next step is to switch to 2D/3D infiltration experiments, which exhibit much more complex behavior compared to 1D experiments. However, despite the development of many models over the last few decades, none of them have

been able to correctly capture 2D/3D laboratory experiments during water infiltration. The most recent and promising attempt was made by Beljadid et al. [9], but the authors observed preferential flow for very low influx instead of diffusion flow.

Again, this point is related to the manuscript motivation and is thus closely tied to all changes performed in the Introduction section. Specifically, this point is discussed on lines 45–54 and 94–105.

- Moreover, we conjecture that 20/30 sand is not an artificial porous medium. In this case, we refer to the response to the reviewer's comment n. 9, where we discuss in detail the porous material used.

This point is discussed below in reviewer's comment n. 9.

Reviewer comment n. 7: The Results are tedious to read.
Too much literature is discussed in the Results section. It slows down the pace and increases the wordiness of the section. The literature review belongs in the Introduction.

- We agree that the Results section is overly extensive, and we plan to shorten this section. Specifically, literature-related content will be predominantly included in the Introduction section. Further details are provided in our annotated PDF.

We have shortened extensive parts of the manuscript as follows. Some of the changes are related to the reviewer's annotated PDF file.

Sect. 1 *Introduction*:

- The paragraph on lines 36–43 from the previous manuscript has been removed, as it was not considered highly relevant to the manuscript.

Sect. 2 *Methods*: All the changes performed in this section, including shortening extensive parts, have been discussed in reviewer comment n. 6. Therefore, we do not repeat them here.

Sect. 3 *Results*:

- Lines 252–260: The first three paragraphs of Sect. 3.1 have been shorten to one paragraph. Literature related content has been moved into the Introduction section.
- Sect. 3.4 *Finger width as function of influx*: This section has been clarified and shortened accordingly.
- Sect. 3.5 *Preferential flow as function of influx*: This section has been significantly shortened, and the text has been clarified accordingly. We have removed the first paragraph and the corresponding Fig. 7, as these are not relevant to the main conclusion of this section. We have also clarified the caption of Fig. 8 (currently Fig. 7).
- Sect. 3.6 *Wetting front instability*: The first paragraph has been removed by moving literature related content to the Introduction section. The last paragraph has also been removed as it was a repetition of the text from the Discussion Sect. 4.2. The section Sect. 3.6 has been clarified accordingly.

Sect. 4 *Discussion*:

- The first paragraph on lines 383–393 from the previous version has been clarified and moved into the Introduction section.
- The text on lines 394–415 from the previous version has been shortened and clarified accordingly. The modified text now forms Sect. 4.1 *Preferential flow in the case of fully wetted porous medium* on lines 396–406.
- The text on lines 416–457 from the previous version has been modified. Some parts from the previous version have been shortened or removed. The modified text now forms Sect. 4.2 *Formation of saturation overshoot and stabilization of wetting front* on lines 408–450.

Reviewer comment n. 8: You only appear to pursue to reproduce experiments in 2D sand tanks under highly idealized conditions that have no relevance for field scale problems related to preferential flow.

- Again, as discussed in the previous responses, e.g., in the response to the reviewer's comment n. 3, we disagree that the idealized conditions used in laboratory experiments are not relevant for in-field experiments. Simply put, any model that struggles to replicate small-scale experiments is inappropriate for addressing field-scale problems.

  This point is related to the changes performed in the Introduction section.

Reviewer comment n. 9: What is the point of developing a model that cannot be applied to address problems in natural and agricultural soils, but is entirely focused on experiments with artificial porous media that were carried out to reveal the physical mechanism driving wetting front instability, so that the science could move on and make this knowledge operational for field problems?

- It is important to note that there are no limitations when using the model in natural and agricultural soils. As we have discussed in our previous responses, we consider it crucial to first validate the model on well-documented small-scale experiments, where details can be accurately captured. Moreover, while laboratory experiments have been carried out to reveal the physical mechanism driving wetting front instability, it is again worth noting that no model has yet been able to fully capture this physical mechanism in detail [18]. Some may argue that capturing the details of flow on a small-scale is unnecessary and that the focus should be on a field-scale. However, it is essential to ensure that any model primarily designed for small-scale simulations should first succeeds at this scale before attempting field-scale simulations. This approach is also consistent with the arguments of DiCarlo [6], who, along with his colleagues such as Steenhuis, Parlange, or Bauters, has conducted a thorough experimental laboratory analysis. This raises the question: why use a model for field-scale problems if it cannot even simulate laboratory experiments with "artificial porous media". How then can such a model be expected to succeed in much more complex situations?

  In principle, approaches to modeling porous media flow can be divided into two categories [19]: (1) microscale models, which are developed to a local scale where the Darcy-Buckingham law can be applied and (2) and macroscale models, which focus directly on the large scale where the Darcy-Buckingham law is not applicable. An interesting model in this regard is the ARM model, based on fractal flow pattern [19]. When using macroscale models, it is not possible to validate the model on small-scale laboratory experiments due to their nature. These models cannot capture fingers because the computational grid blocks are too coarse, so multiple fingers are included within each block. On the contrary, this limitation does not apply for microscale models. However, there is a significant drawback for microscale models: they are very computationally intensive. Since microscale models are developed at a small-scale, it is essential to use a sufficiently refined mesh (below the REV) even for in-field simulations to adequately cover the formed fingers. Therefore, our intention is to move to field-scale simulations (after thorough validation which is necessary), although an alternative discretization approach of the semi-continuum model, such as the Lattice Boltzmann method, needs to be implemented.

  We plan to improve a part of the Introduction section where we discuss different types of modeling (lines 44–64). Specifically, we aim to explain the differences between microscale and macroscale models, focusing on their respective advantages and disadvantages. This is closely related with our motivation for using the semi-continuum model, as previously discussed in our responses to the reviewer's comments n. 1 and n. 3.

  This bullet point is related to the manuscript motivation and is thus closely tied to all changes performed in the Introduction section. Specifically, we have provided a brief overview of microscale and macroscale models, highlighting respective advantages and disadvantages on lines 55–60.

- The hydraulic properties of the 20/30 sand closely match those of the soil cover found in the Czech mountains and the highlands, such as Cambisol developed on Paragneiss (Bohemian Forest, Giant Mts., Bohemian-Moravian Highland). Moreover, the measured Bo number for Cambisol is similar to that of the 20/30 sand [20]. This soil type covers approximately 60 % of the total area of Czechia, with the Bohemian Forest alone representing the water source for approximately 30 % of the country. As a result, systematic monitoring of the water regime in the mountains, including soil water movement, has been carried out for more than 40 years [21, 22]. Therefore, we do not agree that the 20/30 sand is an artificial porous medium; rather, it can be considered a suitable model material for investigating the formation and movement of the wetting front in a real Cambisol.

  The 20/30 sand has also proven to be a suitable model material in research on the oscillation discharge from soil measured in situ in Bohemian Forest [23], as it has been measured in laboratory using this sand [24].

  No changes have been performed regarding this point to avoid excessive extension of the manuscript.

Reviewer comment n. 10: You do not convince the reader why your model matters. Confirming experimental results only serves to show that you model works, so you can apply it to problems beyond experimental reach. Instead, you made it the main point of the paper.

- There are many models that may appear consistent with experiments, but subsequently turn out to be inappropriate overall [18, 25]. This holds true even in cases where the model is consistent with large in-field experiments [11]. Therefore, it is absolutely essential to demonstrate that model works, which is based on the thorough model validation advocated by DiCarlo [6], especially if other models simply cannot do this yet.

  Again, this point is related to the manuscript motivation and is thus closely tied to all changes performed in the Introduction section. An example, that model validation with in-field experiments without proper small-scale validation can lead to incorrect conclusions, can be found on lines 68–78.

Reviewer comment n. 11: Figure out what the real contribution of the paper is, pick the most interesting results, and discuss these intelligently, with an eye to theoretical advances as well as relevance for the real world. Rigorously cut away the obvious stuff. This will require a whole suite of new model runs, new figures, and a more lively discussion.

- We plan to enhance the readability of the text, specifically to shorten extensive parts of the text. Moreover, we plan to provide a clearer explanation of the manuscript's motivation, as previously discussed in our responses. See also our response to the reviewer comment n. 7 for further clarification.

  This has been discussed in previous responses.

Reviewer comment n. 12: You present a state-of-the-art hysteretic model, but use it with outdated soil hydraulic functions. Preferential flow under natural conditions will manifest itself during a season with many cycles of rain and evapotranspiration. This is a wonderful playground for a model that can handle hysteresis, but you have not considered even one cycle. This is one of many missed opportunities. See my comments on the Results Section for more thoughts on this.

- The term "outdated soil hydraulic functions" may not be entirely clear, as we use a standard retention curve defined by the van Genuchten equation along with a relative permeability function. As far as we know, these functions are still commonly used in models [9]. Furthermore, we do not need to perfectly adjust soil hydraulic functions to match the experiments, as we achieved good experimental agreement without doing so.

  According to the reviewer's annotated PDF file, we interpret "outdated soil hydraulic functions" to refer to a retention curve without implementing non-zero water entry values. It is important to stress out that we do not need to use water entry value to initialize the so-called hold-back effect, which subsequently causes the saturation overshoot. For instance, in [26], the flux is directly set to zero when the capillary pressure at the wetting front is higher than the water entry pressure, and non-zero otherwise. This approach creates the hold-back effect and thus the saturation overshoot. Therefore, the creation of the saturation overshoot is prescribed in the model. A similar approach has been undertaken by Steinle et al. [27], where the saturation overshoot was created using a time-dependent Dirichlet condition. We believe that the saturation overshoot should be an output of the model rather than an input. In our semi-continuum model, the hold-back effect is initiated by very low conductivity between blocks if one of them is initially dry. While this is already discussed in the manuscript (page 21, lines 416–428), we plan to enhance this part and move it to the Introduction section for better clarity. We also discuss this point in detail in our annotated PDF (page 21, lines 416–428).

  Let us note that our response regarding the diffusive character of the Richards' equation is also closely related to this point. Please find this response below in the section *Other reviewer's comments*.

  Lines 407–450: We have improved the explanation of the formation of saturation overshoot in the semi-continuum model and how the hold-back effect is initialized in Sect. 4.2 *Formation of saturation overshoot and stabilization of wetting front.*

- The second part of the reviewer's comment concerns the design of the simulation. We agree that this is indeed a wonderful playground, which actually aligns with our future plans, as we assume that the repetition of rain cycles is crucial for understanding the oscillation discharge phenomenon [24]. However, before moving in this direction, an additional step need to be taken. As previously discussed, the model has to be validated in detail. If the model fails to accurately simulate even a laboratory experiment with one cycle of rain, there is no justification for its use in simulating multiple cycles.

  Why the design of the simulations used in the manuscript is relevant has already been discussed in previous responses.

**Other reviewer's comments**

We consider some of the reviewer's comments contained in the annotated PDF file to be quite important, and therefore, we address them below.

Reviewer's comments: What happens if you feed Richards' equation with a retention curve that has a water-entry value at which the hydraulic conductivity exceeds the infiltration rate? If the soil is drier than the water-entry value anywhere, you should be able to reproduce unstable flow with Richards' equation, although there is no guarantee that finger size and spacing will correctly be reproduced. You need to have some heterogeneity in the soil hydraulic properties and the elevation of the soil surface to allow initial instabilities to form. Some of them can then develop to full fingers.

The bottom line is: the problem is not Richards' equation per se, it is the soil hydraulic functions and the initial and boundary conditions that cause it to fail.

As I said before: Richards' equation can if you give it non-zero water entry values. Hysteresis stabilizes a fingered flow pattern [28], but for fingers to form, infiltrating water needs to overcome a threshold. You somehow have not mentioned this at all, so far in the paper.

- It has been mathematically proved that the Richards' equation is unconditionally stable [29] under monotone boundary conditions. The result holds regardless of any particular form of the hydraulic conductivity or the non-decreasing and smooth retention curve, including any type of hysteresis. Therefore, in principle, the Richards' equation cannot admit finger-like solutions in this case. This is not surprising given its parabolic nature. However, if the saturation overshoot is created "manually", the Richards' equation will maintain the overshoot, and hysteresis indeed stabilizes a finger in this case [11, 7]. One of the possibility is to create the overshoot using a time-dependent Dirichlet condition [27] or by defining a bottleneck (zero flux) using a water entry pressure [26]. In our opinion, this is somewhat artificial way to create the overshoot, as the model should ideally be able to generate the overshoot without the need for such ad-hoc threshold directly incorporated into the model. Forming the saturation overshoot should be an output of the model.

  Note that this is also consistent with the arguments of other researchers. For example, consider the work of DiCarlo [13]. Specifically, we refer to paragraph [27] in DiCarlo's manuscript: "In addition, that saturation overshoot occurs in a 1-D system is clear evidence that the 1-D Richards' equation cannot describe the behavior at the wetting front [Eliassi and Glass, 2001] and consequently cannot describe the behavior that causes gravity-driven preferential flow [Jury et al., 2003; Wang et al., 2003]."

  In the manuscript, we plan to specify that the diffusive nature of the Richards' equation is conditional on a smooth and non-decreasing retention curve, as this is not clear from the current text.

  Line 62: We have clarified that the Richards' equation is unconditionally stable with a smooth and non-decreasing retention curve.

  Lines 483–490: We have also included that the Richards' equation can develop saturation overshoot in the case of a time-dependent Dirichlet condition or with a non-smooth retention curve.

Reviewer's comment: That is because the field has moved on to field-scale problems. There, finger sizes and finger spaces do not matter that much. More important is the impact on water availability in the root zone and on leaching of contaminants to the groundwater.

- There are still many recent models attempting to accurately capture small-scale laboratory experiments, typically various extensions of the Richards' equation [15, 26, 9]. We believe that the reason why no model has yet been able to fully reproduce the infiltration dependence in 2D/3D is simply because such a model has not existed yet.

  No changes have been performed regarding this point.

**Proposed revisions to the manuscript**

For the sake of clarity, below is a brief summary of the major proposed changes we plan to make to our manuscript. Minor changes are not listed below, as they are addressed in our annotated PDF file starting from page 9.

All these major changes 1–8 have been incorporated into the manuscript, as discussed in our previous responses.

1. We will clarify the motivation of the manuscript, focusing on the importance of small-scale validation before moving to field-scale simulations.

2. We will provide a brief overview of microscale and macroscale models, highlighting respective advantages and disadvantages.

3. We will improve the explanation of the semi-continuum model with an emphasis on the Prandtl-type hysteresis operator.

4. We will shorten Results and Discussion sections, e.g., by moving the literature-related content to the Introduction section.

5. We will incorporate suggested references into the manuscript.

6. We will discuss the limitations and potential future applications of the semi-continuum model regarding its applicability for field-scale simulations.

7. We will improve the explanation of how the hold-back effect is initialized in the semi-continuum model.

8. We will improve the English.

**Reviewer's annotated PDF file**

Some changes in the manuscript are not mentioned in our previous red-highlighted comments, as the reviewer's comments have been provided also in the annotated PDF file. To provide a complete point-by-point response to all the comments raised by the reviewer, we include a brief list of changes in the manuscript related to reviewer's annotated PDF file. Note that most of the reviewer's comments from the annotated PDF file have already been discussed above; hence, to avoid repetition, these are not included below.

We always include lines on which the reviewer's comment was located in the previous version of the manuscript, along with lines on which changes are implemented in the revised version. Lines referring to the previous version are included in brackets.

- Lines 25–26 (previous: 15–16): The sentence has been reformulated.

- Line 27 (22): The reference has been corrected.

- Line 37 (27): The ordering of references has been fixed.

- Lines 42–43 (33): The sentence has been reformulated by removing a confusing part.

- Lines 52–54 (37): The term "homogeneous soil" has been clarified.

- Lines 108–109, 302 (63): We have clarified that we compare our results with experimentally measured finger widths, not theoretically. Moreover, we have clarified the corresponding text.

- Caption of Fig. 1 (previously caption of Fig. 1): The caption of Fig. 1 has been clarified.

- Lines 123–130 (97–101): This part has been rephrased in Sect. 2.1, where the Prandtl operator is explained in detail.

- Line 251 (191): The subsection has been renamed to Numerical setup.

- Lines 198–202 (previous: the comment in Table 1): We have clarified that using a more complex hysteresis model is not beneficial.

- Line 261 (218): We have clarified that initial and boundary conditions are set to be consistent with the experiments.

- Lines 266–274 (223–227): We have fixed a terminology of the bottom boundary condition and clarified the text accordingly, along with an explanation of residual saturation.

- Line 269 (225): Indexes have been changed from italics to regular font.

- Lines 275–278 (229): The paragraph related to the intrinsic permeability distribution has been reformulated. The term "homogeneous soil" has been clarified on lines 52–54.

- Fig. 10 (previous: Fig. 11): The legend of this figure has been corrected.

- Line 430 (434): The English-related mistake has been fixed.

- Lines 438–441 (442–443): We have acknowledged other studies and the paragraph has been clarified accordingly.

- Line 606 (539): The name has been corrected.

**Reviewer #2**

**Part 2**

Reviewer's comments: `https://doi.org/10.5194/egusphere-2023-2785-RC3`
Author's response: `https://doi.org/10.5194/egusphere-2023-2785-AC3`

Reviewer comment: In the response to one of my comments, the authors reply: "It has been mathematically proved that the Richards' equation is unconditionally stable [29] under monotone boundary conditions. The result holds regardless of any particular form of the hydraulic conductivity or the non-decreasing and smooth retention curve, including any type of hysteresis. Therefore, in principle, the Richards' equation cannot admit finger-like solutions in this case. This is not surprising given its parabolic nature. However, if the saturation overshoot is created "manually", the Richards' equation will maintain the overshoot, and hysteresis indeed stabilizes a finger in this case [11, 5]. One of the possibility is to create the overshoot using a time-dependent Dirichlet condition [27] or by defining a bottleneck (zero flux) using a water entry pressure [26]. In our opinion, this is somewhat artificial way to create the overshoot, as the model should ideally be able to generate the overshoot without the need for such ad-hoc threshold directly incorporated into the model. Forming the saturation overshoot should be an output of the model."

I disagree with the characterization of a water-entry value as an "ad-hoc threshold". I argue instead that the water-entry value is a necessary physical attribute of the soil: not requiring a water-entry value is equivalent to setting the matric potential at which water can enter a dry soil at $-\infty$ (arbitrary units of length, if the matric potential is expressed as energy per unit weight). When the water-entry value is $-\infty$, the Laplace-Young Law stipulates that the following equality holds:

$$r_1^{-1} + r_2^{-1} = \infty$$

where $r_1$ and $r_2$ are the principal radii of curvature of the air-water interface at the pore where water enters the soil at infinite matric potential. This equality can only hold if at least one of these radii equals zero. Such a pore can obviously not exist, and if it could, it would be unable to conduct water.

We agree with the reviewer that the water-entry value is an important physical attribute of the soil. Let us clarify our response, as it was not very well explained. The porous medium comprises many pores, each characterized by a specific water-entry value determined by the Young-Laplace equation based on its principal radii. The shape of the main wetting branch of the retention curve is given by the combination of various pores, hence is a result of the combination of the various pore water-entry values. Consequently, this combination of water-entry values is integrated into the main wetting branch. The pores with the smallest radii determine the lowest pressure $P_{low}$ and the corresponding lowest saturation $S_{low}$, marking the beginning of the main wetting branch. At the point $[S_{low}, P_{low}]$, the main wetting branch is obviously not smooth. However, our initial saturation is significantly higher than the lowest saturation $S_{low}$, hence the non-smoothness of the main wetting branch does not affect the results obtained. Furthermore, the stability proof of the Richards' equation, as derived in [Fürst et al., 2009], remains valid in this scenario. This is because the saturation values lie outside the interval containing $S_{low}$, where the main wetting branch is smooth and non-decreasing.

One might argue that even if the chosen initial saturation is above $S_{low}$, it would still be reasonable to define the value $S_{low}$ in the retention curve, ensuring the model's validity for lower initial saturation. This is clearly not implemented in our model, and the retention curve satisfies $P \to -\infty$ for $S \to 0$. However, in this case, the calculated flux in the semi-continuum model equals zero.

To illustrate, consider two blocks: one fully saturated ($S = 1$) and the second block with saturation decreasing towards zero. The flux between blocks is then given by equation:

$$q = \frac{\kappa}{\mu}\sqrt{k(S_1)k(S_2)}\left(\frac{P_1(S_1) - P_2(S_2)}{\Delta x}\right),$$

where index 1 denotes the fully saturated block, while index 2 denotes the block with the saturation decreasing towards zero. In the equation, $\kappa$ and $\mu$ represent the relative permeability and dynamic viscosity, respectively, and $k(S)$ denotes the relative permeability. The pressure is determined by the van Genuchten's equation and the relative permeability is defined by equation (5) in the manuscript. Note that the fully saturated block satisfies $k(S_1) = 1$ and $P_1(S_1) = 0$. For the sake of simplicity, let's assume $\kappa$, $\mu$ and $\Delta x$ are all equal to one, as these values are independent of saturation and thus does not affect the limiting process. The limit for $S_2 \to 0$ is then simplified to:

$$\lim_{S_2 \to 0} q = \lim_{S_2 \to 0} -\sqrt{k(S_2)}P_2(S_2).$$

In the figure below, the numerical limiting process is depicted using the parameters specified in the manuscript. It is evident from the figure that limit approaches zero, confirming that the flux indeed equals zero for $S_2 \to 0$.

[Figure]

Therefore, the block with zero saturation cannot conduct the water and thus represents a hypothetical pore with zero radii. This is consistent with the reviewer's argument, as he demonstrates that for $P = -\infty$, the Young-Laplace equation yields a pore with zero radii. It's important to note that the limit equals zero only due to the application of the geometric mean of the relative permeability.

If the arithmetic mean is used instead, the limit satisfies:

$$\lim_{S_2 \to 0} q = \lim_{S_2 \to 0} -\frac{1 + k(S_2)}{2}P_2(S_2).$$

Therefore, the limit will approach infinity, as the relative permeability equals $\frac{1}{2}$ for $S_1 = 1$ and $S_2 = 0$. This is well observed in the figure below, where the numerical limiting process using the arithmetic mean is depicted.

[Figure]

Without the application of the geometric mean, unrealistic behavior would occur: the flux would rapidly increase as saturation decreases, which is clearly not physically correct. Additionally, the geometric mean plays a crucial

role in creating the hold-back effect in the semi-continuum model. This effect occurs due to the the very low relative permeability between the blocks, which is a direct consequence of the applied geometric mean.

Finally, we would like to stress out that we find this discussion quite important, especially regarding the behavior as saturation decreases towards zero. Therefore, we plan to include into the manuscript the necessity of the geometric mean in our model in such scenarios and, as a consequence, that blocks with zero saturation represent hypothetical pores with zero radii that are unable to conduct water.

Lines 451–490: This point is extensively discussed in Sect. 4.3 *Importance of the geometric mean in terms of water entry value.* Moreover, the comparison between the geometric and arithmetic mean in terms of calculated horizontal flux is shown in Fig. A6 in Appendix A5.

**Reviewer #2**

**Part 3**

Reviewer's comments: `https://doi.org/10.5194/egusphere-2023-2785-RC4`
Author's response: `https://doi.org/10.5194/egusphere-2023-2785-AC4`

Reviewer comment:
Dear authors,

Apologies for the late reply - field work and a conference interfered.

Just a short response to exppress my appreciation for the insightful response. I find myself in the strange situation that I have learned more from out discussion than from the paper that sparked it. I hope you will find a way to infuse the paper with the thoughts you presented in response to my comments.

Sincerely yours.

- Dear reviewer,

  Thank you for your positive feedback. We also appreciate the discussion that emerged during the review process and find it interesting. We plan to enhance the manuscript based on our previous responses.

  No changes have been performed regarding this point.

**Reviewer #3**

Reviewer's comments: `https://doi.org/10.5194/egusphere-2023-2785-RC5`
Author's response: `https://doi.org/10.5194/egusphere-2023-2785-AC5`

Reviewer comment: I read with great interest this manuscript which reports about the attempt at describing the fingering process in a two dimensional porous medium by means of a semicontinuum approach. The article is rich in the literature and in the analyses. As a general recommendation I suggest:

1. To describe with some more details the background assumptions and the model;

2. To perform a more detailed comparison (even quantitative, if possible) between previous experiments and the numerical findings

In the followings I detail a little more the questions arised from the reading of the paper, that I recommend to address in the review.

- Thank you. Since these general recommendations are discussed in more detail below, we have chosen to comment on them there.

Reviewer comment: The Authors represent a soil section with three main assumptions:

1. The SWRC is represented by a mixed form of a classical van Genuchten's SWRC (in an hysteretic form) with a minimal Prandtl's hysteresis operator. The merge of the two curves is ruled by the dimension of the simulation cell. This idea is close to that the experimental SWRCs are sensitive to the dimension of the laboratory sample, being more flattened as soon as the sample dimension is reduced. In this sense the Prandtl operator is seen as the minimal behaviour of a single capillary. Then they fix the cell dimension and accordingly a SWRC for the soil is found.

   I recommend to better evidence the reason for the choice of the cell dimension, and to express with more detail whether the simulations are expected to be sensitive to the cell dimension (as the SWRC seems conditioned to it).

- The reference block size $\Delta x_0$ of the 20/30 sand was calibrated in [8] by simulating experiments conducted by Bauters et al. [30]. In [8], we ran several simulations of the semi-continuum model using three different values of $\Delta x_0$, and we calculated the moisture profile for three different initial saturations: a dry, a medium dry, and a wet porous medium. We selected $\Delta x_0 = \frac{10}{12}$ as the most appropriate option. Further details of the calibration process are provided in Section 3.1 in [8].

   Here, we used the same parameters of 20/30 sand as given in [8]. This includes parameters such as the van Genuchten's parameters of the retention curve, relative permeability exponent, intrinsic permeability, and the reference block size. Hence, our aim was to use these identical parameters for different flow phenomena, thus avoiding parameter fitting to achieve the best possible agreement with the experiments.

   Lines 252 – 260: The paragraph about the parameters used for simulations has been rephrased to clarify that our intention was not to fit parameters to achieve the best possible agreement with the experiments.

- Since the retention curve is sensitive to the dimension of the laboratory sample, this influences simulations that are sensitive to the size of the reference block $\Delta x_0$. This poses a minor limitation of the semi-continuum model because its objective is to model flow phenomena below the reference elementary volume, where the dependency on the sample volume is significant. However, once the reference block size $\Delta x_0$ is appropriately set, simulations are no longer sensitive to the specific block size $\Delta x$ used. Therefore, the results are independent of the size of the elements used for simulations, which is a crucial characteristic for any numerical model. For example, we refer to Figures 4–6 in [31], where the convergence of the moisture profile in 1D/2D is demonstrated for the block size varying over almost two orders of magnitude.

   Lines 528 – 533: We have addressed this point in Sect. 4.5 *Model validation and future plans*.

- We plan to include an explanation for the choice of the reference block size $\Delta x_0$ to clarify that our aim was not to fit the results. Moreover, the results are not sensitive to the block size $\Delta x$, while this does not apply to the reference block size $\Delta x_0$, which is a parameter of the semi-continuum model.

2. The intrinsic permeability is represented by means of a stocastic simulation. It is really needed? Probably it seems not, as the Authors report in the discussion, but it anyway could mix the effects of hysteresis with the effect of the permeability field on affecting the formation of preferential pathways. Have the Authors considered the possibility of performing separate simulations to evidence the relative importance of hysteresis vs permeability field?

- Since the reviewer repeatedly raised questions about the importance of intrinsic permeability distribution, we decided to comprehensively address all questions at this point. Subsequently, we will extend our explanation in the manuscript of why we used a distribution of intrinsic permeability and what governs the formation of the saturation overshoot.

- The saturation overshoot is formed by two factors: (1) the geometric mean for averaging the permeability and (2) the dependency of the retention curve on the block size, the so-called the scaling of the retention curve.

  The geometric mean plays a crucial role in creating the hold-back effect in the semi-continuum model. This effect occurs due to the the very low relative permeability between the blocks, which is a direct consequence of the applied geometric mean. However, without scaling of the retention curve, the overshoot would disappear as the block size $\Delta x \to 0$. This has already been discussed in [31]; we refer to Figure 2 in [31], where we demonstrated that the saturation overshoot disappears when the scaling of the retention curve is not included. Let us note that the geometric mean can be replaced by any type of averaging that satisfies the crucial property of being small if one of the averaged numbers is small. Hence, it is also possible to use, for instance, the harmonic mean. We chose the geometric mean, as it is more appropriate in the case of a random stratified medium [32], while the harmonic mean is more suitable for a perpendicular stratified medium.

  Lines 411–421: The explanation of saturation overshoot formation is included in Sect. 4.2 *Formation of saturation overshoot and stabilization of wetting front.*

- The formation of the saturation overshoot is not influenced by the distribution of the intrinsic permeability. This is evidenced by 1D simulations; the saturation overshoot is formed even when the distribution is not used, as shown in Figure 10 in the manuscript. Moreover, this corresponds with findings from experimental observations [6], where the formation of saturation overshoot is not determined by heterogeneity. Hence, the question arises: Why use the distribution of the intrinsic permeability if it does not affect the formation of the saturation overshoot? Because a slight distribution of the intrinsic permeability will cause the water to not flow uniformly through the entire porous medium in 2D/3D. This is expected because no heterogeneity is introduced in the governing equation; hence, water will always flow uniformly unless some additional heterogeneity is introduced. This intrinsic permeability distribution is typical for similar models used for unsaturated porous media flow [14]. Another possibility is, for instance, using initial saturation with small perturbations near the top boundary, as used in [9], or introducing small perturbations in a top boundary condition. We plan to include separate simulations in the Appendix without intrinsic permeability distribution to demonstrate that the used distribution is not the cause of the formation of the saturation overshoot in 2D, but rather causes water to flow preferentially.

  To summarize: (1) When using only the distribution of the intrinsic permeability without incorporating the geometric mean and the scaling of the retention curve, the overshoot will not be formed. Consequently, the flow behavior is diffusive in this scenario and water flows throughout the entire porous medium. (2) When using only the geometric mean and the scaling of the retention curve, the overshoot can be formed (depending on initial and boundary conditions). However, the flow remains uniform throughout the entire porous medium. By combining (1) and (2), the saturation overshoot is formed and water does not flow uniformly.

  Moreover, when the distribution is employed but the saturation overshoot does not occur (e.g., in the case of low influx), water tends to flow diffusively in the semi-continuum model. This aligns well with experimental observations. Other similar models fail in this case, as water flows preferentially even when the saturation overshoot is not formed. In our case, if the overshoot is formed, preferential flow is observed. If there is no overshoot, preferential flow disappears. The distribution of the intrinsic permeability is included to make water flow non-uniformly (preferentially) throughout the entire porous medium, but this non-uniformity occurs only when the physics of the semi-continuum model allows it, i.e., when the overshoot is formed. And this applies to any heterogeneity that can be included in the model, such as small perturbations in the initial and boundary conditions.

  Lines 491 – 519: The effect of intrinsic permeability distribution is addressed in Sect. 4.4 *Effect of intrinsic permeability distribution.* Moreover, Fig. A5 in Appendix A4 shows saturation profiles for four different infiltration rates without distribution of intrinsic permeability.

3. The relative permeability is represented by means of a classical power law / or van Genuchten Mualem shape.

With these hypotheses, it is assumed that the conservation of mass with the Darcy–Buckingham law, admits the formation of a saturation overshoot in the 1d form, and the formation of front instability in the 2d form (as it does in the simulations). The obtained equation is in my opinion a form of the Richards equation with hyteretic SWRC, because, as far as the soil is discontinuous and we aim at representing its properties by means of descriptive index properties (as the saturation or the porosity are), we implicitly admit that the Richards equation is locally defined by means of average values on a certain small domain, which is commonly referred to as the Representative Elementary Volume – even if not defining its dimension. What moves the present model from the Richards equation is the choice of the dimension of the cell, which rules the behaviour of the SWRC.

- While it is true that the semi-continuum model is similar to the Richards' equation, it is important to note, as pointed out by the reviewer, that the block size rules the behavior of the retention curve. This makes the formal limit of the semi-continuum model mathematically significantly different. Richards' equation is a parabolic differential equation, whereas the formal limit of the semi-continuum model is a parabolic-hyperbolic equation.

  Lines 244–249: We have slightly modified the paragraph related to the difference between the Richards' equation and our model. Moreover, we have clarified why the formal limit of the semi-continuum model is a hyperbolic-parabolic differential equation in the case of an unsaturated porous medium on lines 149–152. These changes are mostly associated with the comments of reviewer #2, but are also relevant to this comment.

Reviewer comment: Regarding the formation of an overshoot the Authors refer together to two kinds of overshoots which are very different in their meaning. In fact an overshoot related to a pulsation in the Dirichlet boundary condition is in any case in agreement with the parabolic behaviour of the Richards equation, as it can be in any case framed within the maximum principle which guarantees the solution of the parabolic operators. Consider, for example, the Stokes problem of the velocity profile in a seminifinte steady fluid with pulsating wall.

- We agree with the reviewer. For instance, a time-dependent Dirichlet boundary condition could indeed form saturation overshoot in the parabolic differential equation [27]. However, since the experiments we are replicating utilized a monotonic Dirichlet boundary condition, any pulsations in the boundary condition are of no interest. In this case, the Richards' equation with a smooth and non-decreasing retention curve is unconditionally stable [29]. We will clarify this point in the manuscript.

  Line 62: We have clarified that the Richards' equation is unconditionally stable with a smooth and non-decreasing retention curve.

  Lines 483–490: We have also included that the Richards' equation can develop saturation overshoot in the case of a time-dependent Dirichlet condition or with a non-smooth retention curve. This part is again mostly associated with the comment of reviewer #2.

Reviewer comment: Another is the case for which the maximum principle is not proven, as it can happen in some cases in which the soil is not homogeneous (see Barontini et al, 2007, WRR). This is the intriguing case which, according to the Authors, may lead to the instability. Is it possible to check the applicability of the maximum principle for the investigated case? I mean: the maximum principle is unprovable in the equation? or it is provable in the equation but it can be put into discussion as a consequence of the unhomogeneities of the soil permeability?

- As previously discussed, soil permeability inhomogeneities are not the cause of the saturation overshoot. The heterogeneity of the porous medium is neither a necessary nor a sufficient condition for the formation of instabilities.

  Moreover, we did not perform an analysis regarding the maximum principle. The saturation overshoot is not conditional on soil permeability inhomogeneities, hence the maximum principle cannot be proven. It would be interesting to attempt to prove otherwise; that the solution of the semi-continuum model is unstable. However, the formal limit of the semi-continuum model is a rather complex mathematical object, and we are not aware of any research addressing equations of this type. Therefore, such an analysis would not be straightforward. Anyway, we appreciate the suggestion for potential future research.

  The performed changes related to intrinsic permeability distribution have been addressed in previous comments.

Reviewer comment: The results are interesting, particularly as the model describes a concentration of flow in the most saturated soil, even in the case in which the fingers are not developed. This is in agreement with the fact that the relative conductivity drops down as soon as the soil is not completely saturated. This behaviour is most evident in organic soils, where van Gencuchten's n is small (smaller than 2), but it is evident in any porous medium, also in this case for n between 6 and 8.

- This occurs for an intermediate case between the diffusion and finger-like flow, with the fingers not fully developed. This output from the the semi-continuum is interesting for us as well, as it provides detailed insight into the transition from diffusion to finger-like flow.

  No changes have been performed regarding this point.

Reviewer comment: Yet one may argue whether the results account for a physical behavious or for a mathematical description intrinsic to the model. This is why I recommend to the Authors (1) to better focus on which is in their opinion the physical source fo the instability, whether it is the hysteresis or the variability of the permeability. (in this case it cannot be the presence of macropores, as – if I properly understand – macropores are not describerd in the model) and (2) to provide closer comparison with literature experimental results.

- Regarding the physical source of the instability, we refer to our previous responses to avoid repetition. Additionally, it is worth noting that macropores are indeed not included in the model.

  The performed changes related to this point have been addressed in previous comments.

- Regarding a closer comparison with experimental results from the literature, we have already performed several comparisons in 1D and 2D. Specifically, we refer to [17] for 1D simulations and to [7, 8] for 2D simulations. For instance, in [8], we demonstrated that the semi-continuum model is capable to fully reproduce the transition from finger-like flow in an initially dry medium do diffusion-like flow in an initially wet medium for a point source infiltration.

  We plan to better explain our motivation in the manuscript, and why we chose to simulate 2D infiltration experiments of Yao and Hendrickx [5] and Glass et al. [4]. Our goal is to fully validate the semi-continuum model with well-known laboratory experiments, following DiCarlo's approach [6]. DiCarlo suggested evaluating a model, which consists of four points (see section 6.6 in [6]). The only point the semi-continuum model has not fully addressed is the fourth one, which states that the model "Can produce predictions of the 2-D and 3-D preferential flow in terms of finger widths and finger spacings." We have already achieved successful reproduction of the dependency on initial saturation [7, 8]. However, the final aspect required for full validation is to accurately capture the dependence on the infiltration rate of 2D/3D experiments [5, 4, 6]. Furthermore, ongoing efforts within the community, as evidenced by Beljadid et al. [9], emphasize the continued interest in reproducing these experiments.

  The manuscript motivation has been detailed in the Introduction section; a detailed description is provided in the response to the first part of the review from reviewer #2.

  Regarding model validation, we have addressed DiCarlo's approach of model validation on lines 79–93. This paragraph also includes another comparison of the semi-continuum model with experimental results. Furthermore, validation of the semi-continuum model is also discussed in Sect. 4.5 *Model validation and future plans*.

Reviewer comment: Before closing, I add some minimal notes: The Authors express the conductivity of the cell as the geometric average of two conductivities at different water content (the minimum and the maximum, it seems, but it shoud be probably better enlightened). Where does this scheme come from?

- We do not use either minimum or maximum of the geometric mean. The misunderstanding likely arises from the notation $k(S^-)$ and $k(S^+)$ in Eq. (1b). These denote the left and right limits, as described in Eq. (1b), i.e., $S^{\pm}(x_0, t) = \lim_{x \to x_0^{\pm}} S(x, t)$.

  Notation in Eq. (1b) may be confusing, and we intend to improve it. The confusion likely comes from the presence of discontinuous saturation, which arises from the use of a geometric mean conductivity and subsequent limiting process [31]. Due to the complexity caused by the use of a Prandtl-type hysteresis operator, we were unable to prove whether the saturation remains continuous in the limit or not. Therefore, for an accurate mathematical description, it is necessary to preserve the discontinuity when using the geometric mean for conductivity.

 We have clarified in the manuscript text that $k(S^-)$ and $k(S^+)$ denote the left and right limits. Moreover, we have improved the notation of the governing equation and clarified the text according to the comment of reviewer #2.

- We use a geometric mean of conductivity for two adjacent blocks, as described in Eq. (6) in the manuscript. This type of averaging is consistent with the work of Jang et al. [32]. Note that the geometric mean has a desirable property of being small if one of the conductivities is small, which does not apply for the more standard arithmetic mean. This has several practical implications; for instance, it eliminates the necessity to implement the water-entry value in the retention curve, as the geometric mean ensures nearly zero flux for very small saturations. This feature has already been discussed here: `doi.org/10.5194/egusphere-2023-2785-AC3`, and we plan to incorporate this information into the manuscript.

  The importance and practical implications of using the geometric mean are also discussed in Sect. 4.2 (lines 411–421), Sect. 4.3 (lines 451–490), and Sect. 4.4 (lines 508–513).

Reviewer comment: Mualem's parameter $\ell$ is usually set at 0.5 (despite it can be changed, if needed): why did the Authors set at 0.8? Does it come form a fit of an experimetnal conductivity curve?

- We did not fit any experimental conductivity curve but used the same parameters as in [8], including the relative permeability exponent $\lambda$. This decision was made to demonstrate that the model is capable of simulating two different flow phenomena without the need for additional parameter adjustment. Our aim was not to optimize the parameters to achieve the best agreement with experiments, although the agreement is already very good.

  However, it's worth noting that the value of the parameter $\lambda$ is consistent with measurements [33]. We intend to clarify this point in the manuscript.

  Lines 253: We have specified that the value of the parameter $\lambda$ is consistent with measurements.

Reviewer comment: Figure 1 is not very clear, I recomment to add the direction of the potential axis

- We agree that Figure 1 is indeed not clear. The same issue was raised by Reviewer n.2, so we have included a part of our response below for clarity.

- The reader is typically not familiar with a partial differential inequality, which is characterized in the model by a Prandtl-type hysteresis operator. To improve clarity, we have chosen to modify Figure 1 in the manuscript to better illustrate the Prandtl-type hysteresis model. Moreover, given that the Prandtl-type hysteresis operator provided by Eq. (1a) is not widely used in the soil science community, we plan to explain it before introducing the governing equation. Please find the modified Figure 1 of the Prandtl-type hysteresis operator below. In this figure, blue lines represent the limits of main wetting and draining branches, while black lines represent non-vertical scanning curves. The mathematical description of the Prandtl-type hysteresis operator can be found in Visintin [10] on page 16 (eqs. 2.3 - 2.5); see also the corresponding Figure 3 on page 15.

[Figure]

Figure 1: Prandtl-type hysteresis operator.

Lines 117–135: The illustration of the Prandtl-type hysteresis operator has been modified (Figure 1 in the manuscript). Moreover, the Prandtl operator has been explained before introducing the governing equation in Sect. 2.1 *Prandtl-type hysteresis operator*.

Reviewer comment: eq. 3: why 0.5?

- We chose to use the midpoint, but any arbitrary value from the interval $[0, 1]$ can be chosen, e.g., the inflection point. However, the effect on the results is negligible because the flux is calculated relative to the pressure gradient. We plan to clarify this in the manuscript.

  Lines 185–187: This point has been clarified.

Reviewer comment: l.226: please better detail the meaning of residual and initial saturation as residual is greater than initial

- The residual saturation refers to the maximum amount of water the porous medium can retain against the force of the gravity. The terminology is borrowed from Bauters et al. [30], where the authors filled the chamber of the 20/30 sand with water and then drained it to a residual water content of $0.047\,\mathrm{cm}^3/\mathrm{cm}^3$. Therefore, in our simulations, we have chosen the residual saturation to be 0.05. Initial saturation, on the other hand, refers to the saturation at the beginning of the simulation.

  Lines 270–272: The meaning of residual saturation has been addressed.

Reviewer comment: How the bottom boundary condition was chosen? If the column is suspended a seepage face condition would have been more realistic, yet it requires soil saturation before the leakage starts;

- Firstly, let us stress that the terminology used in the manuscript is incorrect. We describe the bottom boundary condition as a "free discharge"; however, free discharge actually refers to unit gradient flow at the lower boundary. We will fix this terminology issue in the manuscript.

  Lines 272–274: The terminology has been fixed.

- Secondly, we model the outflow of water into the air. There are various bottom boundary conditions that can be implemented. However, we chose a boundary condition that does not affect the flow above the boundary. Simply put, we wanted to ensure that the boundary condition did not propagate into the porous medium.

  Lines 266–267: We have addressed that we model the outflow of water into the air and that the used implementation does not affect the flow above the bottom boundary.

Reviewer comment: l.443: it is referred to stabilized flow as a percolation flow (i.e. with null spatial rate of change of the tensiometer–pressure potential), but due to the boundary conditions it might not be a pure percolation flow. Thank you for the attention.

- The reviewer n.2 raised a similar question, challenging our statetement: "The saturation and pressure in the finger tail are constant", and refered to the work of Cho et al. [34]. We provide our response to the reviewer n.2 below.

- According to a comprehensive experimental work of DiCarlo [13], the saturation is constant at the finger tail for the uniform top boundary condition. For higher fluxes, slight oscillations are observed, however, DiCarlo explains that these are experimental artifacts related to light transmission variations near the end of the tubes. The constant pressure in the finger tail was also measured experimentally [18].

  For completeness, we will acknowledge studies that have shown the opposite, such as the work of Cho et al. We will also provide references to DiCarlo's research to support our claim.

  Lines 438–441: We have clarified this point.

**Reviewer #1**

Reviewer's comments: `https://doi.org/10.5194/egusphere-2023-2785-RC1`
Author's response: `https://doi.org/10.5194/egusphere-2023-2785-AC1`

Reviewer comment: The manuscript deals with a classical 1D analysis of the unsaturated flow stability during a wetting event.

- It is important to note that the manuscript is not focused on 1D analysis, but 2D analysis instead. Specifically, we show how 2D flow varies with different inflows changing from diffusion flow to fingering flow and back to diffusion flow as the infiltration rate increases (viz results sections 3.1 – 3.5 and discussion section 4).

  In the case of 1D flow, there exist a few models that are able to capture the transition between diffusion and finger-like behavior for various infiltration rates, for instance [14]. However, the 2D case is significantly different because not only this transition but also finger spacing and finger width are addressed [18]. This makes the analysis much more complex and complicated. One of the main evaluating factors according to DiCarlo is that the model "can produce predictions of the 2-D and 3-D preferential flow in terms of finger widths and finger spacings." In our manuscript, we have demonstrated that the semi-continuum model is capable of this. Moreover, to the best of our knowledge, no such model has yet been available to simulate this complex 2D behavior.

  We have improved the motivation of the manuscript in the Introduction section, for example, by incorporating model evaluation suggested by DiCarlo. More details are provided in the response to reviewer #2.

  Let us note that all simulations performed in the manuscript are in 2D. The exception is section *3.6 Wetting front instability*, where we analyze the connection between 1D and 2D flow in unsaturated porous media. This section is included to demonstrate that we are consistent with experimental observations, so that the saturation overshoot in 1D is closely related to the preferential flow in 2D. Again, to the best of our knowledge, such a connection has not been demonstrated by any model so far.

  To make this as clear as possible, we will highlight it in the manuscript in more detail. And we also plan to change the title of the manuscript to make it clear that we are dealing with 2D simulations.

  Lines 3, 18–21: We have clarified that diffusion-like flow refers to stable flow, and finger-like/preferential flow refers to unstable flow.

  We have slightly changed the title of the manuscript to: "Modeling 2D gravity-driven flow in unsaturated porous media for different infiltration rates".

Reviewer comment: The numerical treatment of the problem at stake, although quite standard and therefore with no new insight, appears to be correct (I went through it, and I didn't find any error).

- The numerical discretization of the governing equation is indeed standard with one major exception. The discretization of the Prandtl-type hysteresis operator is not common in flow modeling as the discretization is dependent on the size of the used mesh (blocks). However, this is not a key part of the manuscript as the semi-continuum model has been already published, e.g., in [31] and [8].

Reviewer comment: Instead, the English usage requires a very solid and sounding proofreading.

- English will be proofread for the revised manuscript.

  The English has been revised by the authors and subsequently proofread by an English speaker.

Reviewer comment: Besides this (marginal) aspect, the main issue which I see with the manus is of methodological nature. In particular, my skepticism is two-fold. First, accounting for the gravity solely (thus neglecting the impact of retention) may be unrealistic especially if one is interested (as it is usually happens in the applications) on the "onset" of the stability vsinstability".

- The semi-continuum model describes the flow of water in unsaturated porous medium. Water is characterized by saturation in the equation, since saturation is defined as the ratio of the total water volume to the pore volume. This is a standard approach in modeling multiphase flow [18].

  Note that the gravity is essential in forming the finger flow and is therefore included in the semi-continuum model. However, retention is not neglected at all; the water retention is controlled mostly by the pressure-saturation relation (the water retention curve) defined by equations Eq. (2) and Eq. (3). Moreover, according

to Eq. (6), the gravity term takes a role only in the case of vertical fluxes, while the term related to the retention curve is included for horizontal fluxes as well. Again, this approach is standard to model water retention in porous medium, for example using the Richards' equation [35].

No changes have been performed regarding this point, as we have already explained in the response above that the retention is not neglected.

Reviewer comment: Second, the authors have carried out a long and intensive analysis of the flow rates which make (or not) stable the flow, but what and where is the stability analysis?

- There is probably a misunderstanding in the used terminology and we apologize for that. The stability refers to diffusion-like flow where no saturation overshoot is observed, and the instability refers to preferential flow (or finger-like flow) that is accompanied by the saturation overshoot. This is standard terminology used in the community [6]. However, to make it as clear as possible, we will explain it the manuscript and change the text accordingly. Also, we plan to change the title of the manuscript to: *Modeling 2D gravity-driven flow in unsaturated porous media for different infiltration rates.*

  A short part of the manuscript deals with the instability of the wetting front (section *3.6 Wetting front instability*). As mentioned in the response to the first reviewer's comment, in this section we analyze the connection between saturation overshoot in 1D and preferential flow in 2D. However, this is not the main part of the manuscript because we focus on 2D preferential flow and show that the semi-continuum model correctly captures flow experiments that no model has captured before.

  We have already addressed the same comment in the first response to this reviewer.

Reviewer comment: The very new and innovative insight could have been a Touring analysis of the stable/unstable flow patterns in order to highlight which ones are those parameters (and perhaps the infiltration rate is the most important one) that regulate such a stability. Instead, the manuscript, as it is, is nothing more a numerical analyses (followed by an experimental benchmark), quite similar to many others, already existing in the literature.

- Thank you for your suggestion to include analysis using Turing instability and Turing pattern concept. However, the manuscript addresses different topic as the stability/instability has different meaning (see the response to the previous comment). It deals with 2D preferential flow as a function of applied influx. Specifically, we show that the model is able to predict 2D preferential flow in terms of finger widths and finger spacings.

  According to "quite similar to many others, already existing in the literature": We conjecture that there are no other models that are able to correctly capture the transition from diffusion to finger-like and back to diffusion flow as the infiltration flux increases. In our humble opinion, the most promising model is a nonlocal model proposed by Beljadid et al. [9], however, the flow is preferential for very low fluxes which is not consistent with experiments. This is not the case for the semi-continuum model presented in our manuscript. The model correctly captures the influx dependence even for very low fluxes and, in addition, the finger widths and finger spacings are consistent with experimental observations, as shown in section *3.4 Finger width as function of influx.*

  However, if there is another model dealing with the same topic (i.e., 2D preferential flow as a function of applied influx), we would like to ask the reviewer for a reference, where this issue is properly addressed. We would very much appreciate that.

  No changes have been performed regarding this point.

**References**

[1] M. Kutílek and D. Nielsen. *Soil Hydrology*. Catena Verlag, Germany, 1994.

[2] Milena Císlerová, Jiří Šimůnek, and Tomáš Vogel. Changes of steady-state infiltration rates in recurrent ponding infiltration experiments. *Journal of Hydrology*, 104(1):1–16, 1988.

[3] M. Císlerová, T. Vogel, J. Votrubová, and A. Robovská. *Searching Below Thresholds: Tracing the Origins of Preferential Flow within Undisturbed Soil Samples*, pages 265–274. American Geophysical Union (AGU), 2002.

[4] R. J. Glass, J.-Y. Parlange, and T. S. Steenhuis. Wetting front instability. 2. experimental determination of relationships between system parameters and two-dimensional unstable flow field behavior in initially dry porous media. *Water Resour. Res.*, 25(6):1195–1207, 1989.

[5] T. Yao and J. M. H. Hendrickx. Stability of wetting fronts in dry homogeneous soils under low infiltration rates. *Soil Sci. Soc. Am. J.*, 60:20–28, 1996.

[6] D. A. DiCarlo. Stability of gravity-driven multiphase flow in porous media: 40 years of advancements. *Water Resour. Res.*, 49:4531–4544, 2013.

[7] J. Kmec, T. Fürst, R. Vodák, and M. Šír. A two dimensional semi-continuum model to explain wetting front instability in porous media. *Scient. Rep.*, 11:3223, 2021.

[8] J. Kmec, M. Šír, T. Fürst, and R. Vodák. Semi-continuum modeling of unsaturated porous media flow to explain bauters' paradox. *Hydrol. E. Sys. Sci.*, 27(6):1279–1300, 2023.

[9] A. Beljadid, L. Cueto-Felgueroso, and R. Juanes. A continuum model of unstable infiltration in porous media endowed with an entropy function. *Adv. Water Resour.*, 144:103684, 2020.

[10] A. Visintin. *Differential models of hysteresis*. New York: Springer, 1993.

[11] M. Tesař, M. Šír, J. Pražák, and L. Lichner. Instability driven flow and runoff formation in a small catchment. *Geologica Acta*, 2(1):147–156, 2004.

[12] M. Šír, L. Lichner, J. Kmec, , T. Fürst, and R. Vodák. Measurement of saturation overshoot under grass cover. *Biologia*, 75:841–849, 2020.

[13] D. A. DiCarlo. Experimental measurements of saturation overshoot on infiltration. *Water Resour. Res.*, 40(4):W04215, 2004.

[14] L. Cueto-Felgueroso and R. Juanes. A phase field model of unsaturated flow. *Water Resour. Res.*, 45(10):W10409, 2009.

[15] S. Fritz. *Experimental investigations of water infiltration into unsaturated soil – analysis of dynamic capillarity effects*. Diploma Thesis, University of Stuttgart, Germanys, 2012.

[16] N. Brindt and R. Wallach. The moving-boundary approach for modeling gravity-driven stable and unstable flow in soil. *Water Resour. Res.*, 53(1):344–360, 2017.

[17] J. Kmec, T. Fürst, R. Vodák, and M. Šír. A semi-continuum model of saturation overshoot in one dimensional unsaturated porous media flow. *Scient. Rep.*, 9:8390, 2019.

[18] D. A. DiCarlo. Can continuum extensions to multiphase flow models describe preferential flow? *Vadose Zone J.*, 9(2):268–277, 2010.

[19] H-H. Liu. The large-scale hydraulic conductivity for gravitational fingering flow in unsaturated homogenous porous media: A review and further discussion. *Water*, 14(22), 2022.

[20] Dani Or. Scaling of capillary, gravity and viscous forces affecting flow morphology in unsaturated porous media. *Advances in Water Resources*, 31(9):1129–1136, 2008.

[21] M. Tesař, M. Šír, O. Syrovátka, J. Pražák, L. Lichner, and F. Kubík. Soil water regime in head waterregions – observation, assessment and modelling. *J. Hydrol. Hydromech.*, 49:355–375, 2001.

[22] J. Procházka, J. Pokorný, A. Vácha, K. Novotná, and M. Kobesová. Land cover effect on water discharge, matter losses and surface temperature: results of 20 years monitoring in the Šumava mts. *Ecological Engineering*, 127:220–234, 2019.

[23] M. Šír, L. Lichner, and O. Syrovátka. In-situ measurement of oscillation phenomena in gravity-driven drainage. *In: Elias, V., Littlewood, I.G. (eds.): Proc. Int. Conf. Catchment hydrological and biochemical processes in changing environment, Liblice 1998. Technical Documents in Hydrolog.*, 37:249–255, 2000.

[24] J. Pražák, M. Šír, F. Kubík, J. Tywoniak, and C. Zarcone. Oscillation phenomena in gravity-driven drainage in coarse porous media. *Water Res. Research*, 28(7):1849–1855, 1992.

[25] B. Zhao et al. Comprehensive comparison of pore-scale models for multiphase flow in porous media. *Proceedings of the National Academy of Sciences*, 116(28):13799–13806, 2019.

[26] N. Brindt and R. Wallach. The moving-boundary approach for modeling 2D gravity-driven stable and unstable flow in partially wettable soils. *Water Resour. Res.*, 56(5):e2019WR025772, 2020.

[27] R. Steinle and R. Hilfer. Hysteresis in relative permeabilities suffices for propagation of saturation overshoot: A quantitative comparison with experiment. *Phys. Rev. E*, 95:043112, 2017.

[28] R. J. Glass, J.-Y. Parlange, and T. S. Steenhuis. Mechanism for finger persistence in homogenous unsaturated, porous media: Theory and verification. *Soil Sci.*, 148(1):60–70, 1989.

[29] T. Fürst, R. Vodák, M. Šír, and M. Bíl. On the incompatibility of Richards' equation and finger-like infiltration in unsaturated homogeneous porous media. *Water Resour. Res.*, 45(3):W03408, 2009.

[30] T. W. J. Bauters, D. A. DiCarlo, T. Steenhuis, and J.-Y. Parlange. Soil water content dependent wetting front characteristics in sands. *J. Hydrol.*, 231-232:244–254, 2000.

[31] R. Vodák, T. Fürst, M. Šír, and J. Kmec. The difference between semi-continuum model and richards' equation for unsaturated porous media flow. *Scient. Rep.*, 12:7650, 2022.

[32] J. Jang, G. A Narsilio, and J. C. Santamarina. Hydraulic conductivity in spatially varying media - a pore-scale investigation. *Geophysical J. International*, 184:1167–1179, 2011.

[33] M. Schaap and F. Leij. Improved prediction of unsaturated hydraulic conductivity with the mualem-van genuchten model. *Soil Science Society of America Journal*, 64:843–851, 2000.

[34] H. Cho, G. H. de Rooij, and M. Inoue. The pressure head regime in the induction zone during unstable nonponding infiltration: Theory and experiments. *Vadose Zone Journal*, 4(4):908–914, 2005.

[35] J. Šimůnek and D. L. Suarez. Two-dimensional transport model for variably saturated porous media with major ion chemistry. *Water Resour. Res.*, 30:1115–1133, 1994.

---

## Author Response (AR2)

Dear editor and reviewers, please find below a point-by-point response to the raised comments. Comments from reviewers are highlighted in blue. The format of the colored version of the revised manuscript is as follows: Text that has been added, such as content moved from one section to another, is highlighted in blue, while text that has been removed is highlighted in red.

Before addressing the individual points, we would like to thank both reviewers for their valuable comments during the review process. We believe their comments greatly contributed to improving the manuscript.

Please note that we have updated the references for the simulation data uploaded on Zenodo. Specifically, we have added data for simulations without intrinsic permeability distribution and updated the README file accordingly.

**Reviewer 3**

I read this new version of the article and the Authors replies to my previous notes. The Authors satisfactorily detailed all the issues I raised and I think that the article can be now recommended for publication.
The section 4.4 and the new Figure A5 are useful because provide a clear discussion on which is the relative effect of the distribution of intrinsic permeability, hysteresis and semicontinuum model. It is interesting the fact that the formation of a saturated wetting front is reached, in the semicontinuum model, for q smaller than the conductvity at saturation Ks = 15 cm / min (Figure A2), possibly as a consequence of the saturation overshoot which is not represented by a classical Richards framework.

- We have added a brief comment in section *Dependence of flow on infiltration rate* regarding the formation of a saturated wetting front for infiltration lower than the saturated conductivity, as we find this detail also relevant.

**Reviewer 2**

In the following, line numbers refer to the manuscript with tracked changes.

The new edits introduce considerable numbers of citations after the Introduction (case in point: l. 200-201). This indicates that the Introduction is not well-aligned with the rest of the paper. This is illustrated for the same case by l. 201-202, where you argue why you did not use other hysteresis models. The Methods section is there to detail the methods you used, not to discuss alternatives, that should have been clarified before. There are other examples of this that can be easily fixed by repositioning some of the text in the appropriate section, but I cannot go through all of them. For instance, I believe that the final paragraph of 2.4 contains hardly any methodology at all, but discussed the literature. Therefore, most of it belongs in the Introduction. In the Results and Discussion, more citations occur because you compare your results to those of others. But if you introduce many new papers that were not treated in the Introduction, the Introduction probably can be improved. I did not check this.

- We agree that a considerable number of new citations have been added to Sect. 2.4, particularly in the first and last three paragraphs. To address this, we have moved these parts before the Methods into a new section, Sect. 2, titled *Retention curve and its sample size dependence*. The corresponding text has been slightly modified for clarity.

- We agree that a few citations have been introduced in the Results and Discussion sections. However, we believe that maintaining the flow of the text in these sections is essential for clarity and readability. Therefore, no changes have been made in these sections.

In a few sections of the discussion, you rely heavily on figures in the appendix. Should some elements of the appendix therefore perhaps be moved to the main text?

- We did not want to extend the manuscript too much, however, we agree with this comment. For better continuity, we have moved two figures A5 and A6 to the main text. Consequently, sections A4 and A5 have been removed.

The name 'Rooij' should be replaced by 'de Rooij' throughout.

- We have fixed this issue.

l. 24-25: it is either 'the Richards equation' or 'Richards' equation'.

- We have fixed this issue.

l. 45: Glass et al. (1989c) built a two-dimensional...

- We have clarified the sentence.

l. 120: insert space after the 'for all' operator

- We have fixed this issue.

l. 141, Table 1, possibly elsewhere: Pas → Pa

- Unit of dynamic viscosity is Pa s, not Pa. For clarity, we have added a small space between Pa and s and elsewhere.

l. 144: The gas pressure is not assumed to be zero, but rather is this pressure used as the reference with respect to which the liquid pressure Is expressed.

- We have clarified the sentence.

l. 260: ...$10^5$ Pa, so we set $K_{PS}$ to this value.

- We have clarified the sentence.

Table 1, possibly elsewhere: there need to be spaces between units: $m\,s^{-2}$ instead of $ms^{-2}$

- We have added a small space between units everywhere.

l. 266-267: the boundary condition essentially switches from a no-flow boundary condition (BC) when the matric potential at the bottom is negative to a prescribed flux BC when the pressure potential $>= 0$. In the first case, the BC necessarily affects the region above he boundary because water piles up until the matrix potential at the bottom reaches zero. But why is that a problem?

- Although in the reality, the bottom boundary condition can affect the region above this boundary, we have chosen the implementation given by Eq. (8). This allows us to isolate the behavior of the model from the influence of the bottom boundary condition. Simply put, we want to ensure that the behavior we observe in simulations comes from the model itself and is not caused by a boundary condition. We have clarified this in the corresponding paragraph.

l. 275: What is a small distribution? What are the properties of this distribution?

- We have clarified the corresponding paragraph in the manuscript. For more details, we also refer to [1], specifically pages 75–76.

l. 280 and elsewhere: add a space between a value and its unit(s).

- There actually is a small space between a value and its unit. We have also checked other values in the manuscript.

Fig. 9 and the paragraph above it: There seems to be a discrepancy between the cases shown in the graph and those discussed in the text, or am I overlooking something?

- Thank you for your notice, as it is relevant for the main text of the manuscript. There were two cases where we wrote $q_{\mathrm{top}} = 0.0005\,\mathrm{cm\,min}^{-1}$ instead of $q_{\mathrm{top}} = 0.005\,\mathrm{cm\,min}^{-1}$. We have fixed this issue. Moreover, we have slightly changed the paragraph above Fig. 9.

l. 453: The contact angle of air-water interface with the solid phase is also important. This angle is flux-dependent.

- In a static case, the contact angle of air-water interface is independent of the flux. Since the semi-continuum model uses the retention curve in an equilibrium state, we do not consider this factor relevant for our study. However, it is worth noting that other models directly incorporate the role of the contact angle into the pressure-saturation relationship, such as [2], which is referenced in the manuscript.

l. 494-495: repeats earlier text.

- We have removed the repetition of the earlier text.

**References**

[1] J. Kmec. *Analysis of the mathematical models for unsaturated porous media flow, Ph.D. thesis.* Palacký University in Olomouc, Czech Republic, 2021.

[2] N. Brindt and R. Wallach. The moving-boundary approach for modeling 2D gravity-driven stable and unstable flow in partially wettable soils. *Water Resour. Res.*, 56(5):e2019WR025772, 2020.